# Subtleties in Clathrin heavy chain binding boxes provide selectivity among adaptor proteins of budding yeast

Lucas A. Defelipe [1,2], Katharina Veith[1,2], Osvaldo Burastero [1,2], Tatiana Kupriianova [1,2], Isabel Bento[1], Michal Skruzny[3,4,9], Knut Kölbel[2,5,6], Charlotte Uetrecht [2,5,6,7], Roland Thuenauer[2,5,8] & Maria M. García-Alai [1,2] ✉

Clathrin forms a triskelion, or three-legged, network that regulates cellular processes by facilitating cargo internalization and trafficking in eukaryotes. Its N-terminal domain is crucial for interacting with adaptor proteins, which link clathrin to the membrane and engage with specific cargo. The N-terminal domain contains up to four adaptor-binding sites, though their role in preferential occupancy by adaptor proteins remains unclear. In this study, we examine the binding hierarchy of adaptors for clathrin, using integrative biophysical and structural approaches, along with in vivo functional experiments. We find that yeast epsin Ent5 has the highest affinity for clathrin, highlighting its key role in cellular trafficking. Epsins Ent1 and Ent2, crucial for endocytosis but thought to have redundant functions, show distinct binding patterns. Ent1 exhibits stronger interactions with clathrin than Ent2, suggesting a functional divergence toward actin binding. These results offer molecular insights into adaptor protein selectivity, suggesting they competitively bind clathrin while also targeting three different clathrin sites.

Clathrin-mediated endocytosis is a fundamental and conserved cellular process in eukaryotes that allows cells to internalize cargo like membrane proteins[1–3], nutrients[4,5], and even pathogens[6–10]. It is a concerted process controlled with very regular timing, occurring between 60 s and 120 s from the recruitment of the early adaptor proteins to the membrane and the excision of a vesicle[11]. The protein that gives the process its name, Clathrin, is a triskelion composed of three heavy chains (Clathrin Heavy Chain, Chc) that provide the structural backbone of the lattice, further stabilized and regulated by three light chains (Clathrin Light Chain, Clc). This triskelion is the basic evolutionary conserved structure that forms cages of different sizes and shapes in eukaryotic cells[12]. Depending on the organism, the main role of the clathrin scaffold varies. While in mammals clathrin is

essential for endocytosis[13], in *S. cerevisiae* its main function is controlling the shape and size of endocytic vesicles[14]. Although the *Δchc* mutant of *S.cerevisiae is* viable, it grows slower and shows a reduction in effective endocytic events[15].

Chcs undergo oligomerization mediated by their C-terminal trimerization domains and the protein contains 145-residue repeats known as Clathrin Heavy Chain Repeats. These repeats fold into elongated right-handed superhelical structures composed of short alpha-helices. This structural formation gives rise to several distinct domains within the clathrin molecule: the proximal, knee, distal, and ankle domains[16]. The proximal domain specifically interacts with Clcs, whereas, the N-terminal domain (NTD) of Chc, which adopts a WD40 beta-propeller structure[17], is pivotal for interactions with various

[1]European Molecular Biology Laboratory - Hamburg Unit, Hamburg, Germany. [2]Centre for Structural Systems Biology, Hamburg, Germany. [3]Cell Biology and Biophysics Unit, European Molecular Biology Laboratory, Heidelberg, Germany. [4]Department of Systems and Synthetic Microbiology, Max Planck Institute for Terrestrial Microbiology, Marburg, Germany. [5]Leibniz Institute of Virology (LIV), Hamburg, Germany. [6]Deutsches Elektronen Synchrotron - DESY, Hamburg, Germany. [7]Institute of Chemistry and Metabolomics, University of Lübeck, Lübeck, Germany. [8]Technology Platform Light Microscopy (TPLM), Universität Hamburg (UHH), Hamburg, Germany. [9]Present address: Carl Zeiss Microscopy GmbH, Jena, Germany. ✉e-mail: maria.garcia@embl-hamburg.de

adaptor proteins (APs). APs create a physical link between the clathrin scaffold and the membrane, regulating its invagination and excision. In higher eukaryotes, NTD is known to have multiple binding sites for APs, often called 'boxes'. The first box described by structural studies was the Clathrin box, which was crystallized with peptides from β-arrestin-1 and AP-3 β3a with the canonical mammalian motif (LΦXΦ[D/E], where Φ is any hydrophobic residue). It contains a hydrophobic pocket which binds Leu, Ile, or Phe residues[17] from peptides adopting an extended conformation interacting through hydrogen bonds with a conserved glutamine (Q89). A second site, the W-box, harbours the WxxW motif of Amphiphysin. This peptide folds into a 3–10 helix, which binds to the cavity formed by WD40 repeats[18]. The third binding site is the Arrestin box, which binds (L/I)₂GxL motifs from a splice variant of Arrestin[19]. In this context, the motif is found within a loop and adopts a different orientation compared to the one observed for other peptides binding the Arrestin box.

Studies by Willox and Royle have shown that only when the NTD is removed, a reduction in Transferrin uptake is observed[20]. Following this observation, they have described a fourth binding site within the beta-propeller called the Royle Box. Mutation of the residues belonging to this site (N296A, R297E and E11K), in conjunction with the other three boxes, results in a similar phenotype as deleting the NTD[20].

NMR studies have revealed that a single peptide containing a Clathrin Binding Motif (CBM) is capable of binding to all three NTD boxes of bovine Clathrin[21]. Muenzner's research extends these findings by presenting crystallographic evidence of the *Bos taurus* Chc-NTD in complex with both AP-2 β2 and Amphiphysin CBMs[22], indicating that NTD has dual binding capabilities for Clathrin and Arrestin boxes. However, the question of whether these NTD boxes have developed target-specific binding capabilities remains unresolved.

In yeast, multiple Adaptor Proteins (APs) including epsins Ent1, Ent2, Ent3, and Ent5 and CALM/AP180 homologues Yap1801 and Yap1802 are retained in duplicated form for endocytosis and trafficking functions. These highly homologous protein paralogs have been described to possess redundant functions[23–27]. All these proteins contain short linear motifs (SLiMs) located in intrinsically disordered regions (IDRs) that serve as CBMs.

Ent1 and Ent2 are APs with modular domain architecture, displaying an epsin N-terminal homology (ENTH) domain followed by unstructured regions containing several interaction motifs: two Ubiquitin Interaction Motifs (UIMs), two Asn-Pro-Phe motifs (NPF) responsible for interacting with EH (Eps-15 Homology) domains of Ede1, and a C-terminal CBM. The ENTH domain binds to Phosphatidylinositol 4,5-bisphosphate (PIP₂)[28–32], one of the main signal lipids produced at early stages of endocytosis[33]. The epsin N-terminal homology (ENTH) domain and the AP180 N-terminal homology (ANTH) domain from epsins and Hip1R proteins form a membrane remodelling complex (AENTH) in the presence of PIP₂[34] enabling the connection between the membrane, the clathrin coat and the actin cytoskeleton[30]. Studies by our group have shown that both the ENTH domains of Ent1 (ENTH1) and Ent2 (ENTH2) can form the AENTH complex with yeast Hip1R homologue Sla2[28] coordinating Ent1 and Ent2 at the endocytic site[28]. In addition, it has been described that ENTH is involved in Cdc42 regulation[24] with ENTH2 playing a signalling role during cell division[35]. While most of the research focuses on the membrane binding domains of epsins, experiments comparing functional differences of IDRs involved in establishing the clathrin-adaptors network have not been performed. In this work, we use a combination of structural, biophysical, and live-cell imaging tools to shed light on the interactions between yeast adaptor proteins and Chc. We present crystal structures of CBM motifs from different adaptor proteins bound to the three Chc NTD boxes and suggest a hierarchy of interactions based on in vitro measured apparent dissociation constants. We dissect each clathrin box by mutagenesis to track their occupancy by the Ent1 and Ent2 adaptors and show that both Ent1 and Ent2 exhibit different Chc binding patterns, as revealed by native MS and photobleaching FRET assays. Our experiments suggest that Ent1 displays stronger interactions with clathrin compared to Ent2, which has been confirmed by TIRF analysis.

## Results

### Chc binding motifs in the yeast proteome

To determine which yeast proteins contain Clathrin binding motifs (CBM), we used an iterative search process using Literature search, ELM[36] and SLiMSearch4[37] (see Methods). The final motif found in the yeast proteome is defined as [TSNV]L[IL]D[ILMFW] showing similarities to the mammalian CBM[38] but different in the +3 position; where we find an Aspartate instead of any residue (Fig. 1a). As expected, most of the proteins that contain a CBM are involved in either endocytosis or trafficking processes (Fig. 1c). Among the endocytic proteins, we find adaptors from the early coat (Yap1801 and Yap1802), middle coat

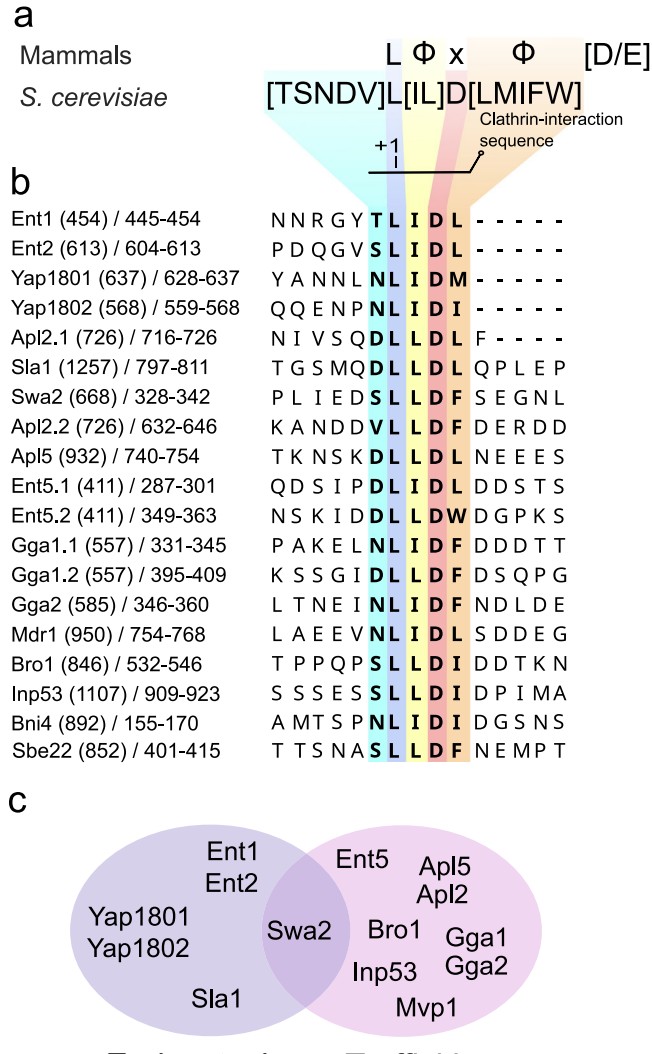

**Fig. 1 | Discovery of a conserved clathrin binding motif in yeast. a** Comparison of mammalian and *S. cerevisiae* CBM motifs. In total, 28 hits in 24 proteins were found in the *Saccharomyces cerevisiae* genome, Φ denotes bulky hydrophobic residues **b** Alignment of CBM motifs from proteins related to endocytosis and trafficking. The numbers in parentheses represent the total protein length, while the numbers following the slash indicate the specific region being displayed. Dots following the protein name denote the sequence instance. **c** Venn diagram of proteins involved in endocytosis (purple) and trafficking (pink) that contain the discovered motif.

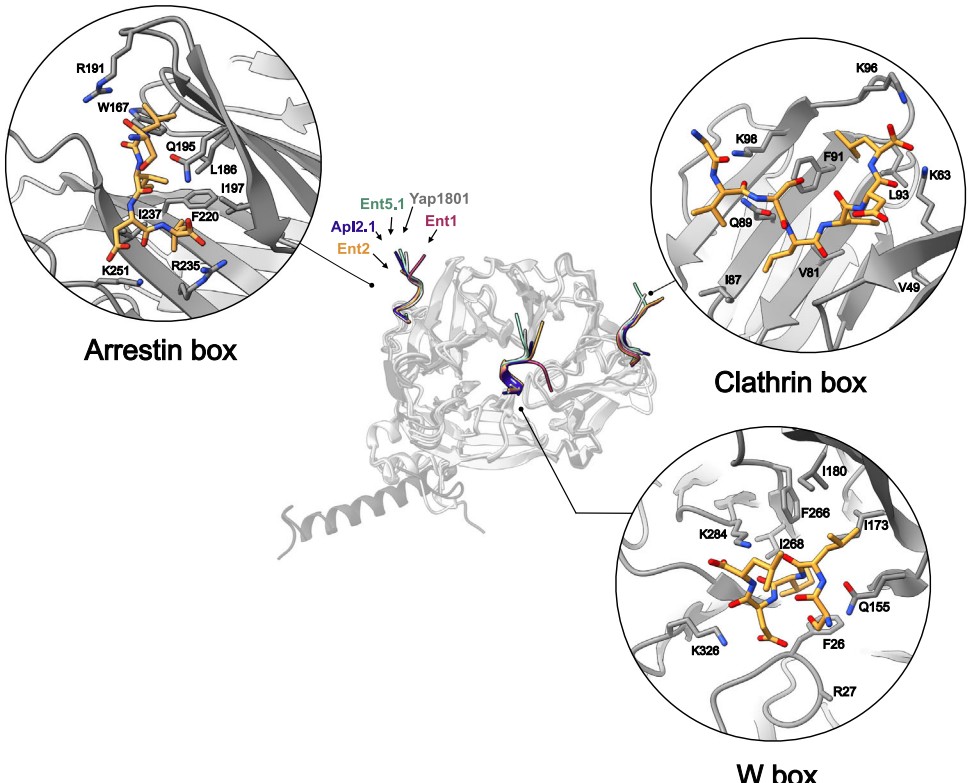

**Fig. 2 | The same CBM motif binds to three different Chc NTD binding sites.** Crystal structure of Chc-NTD (displayed as a grey cartoon) bound to CBMs from Ent1 (1.75 Å, red), Ent2 (1.74 Å, yellow), Ent5 (Ent5.1 LIDL motif, 2.01 Å, green), Apl2.1 (Apl2.1 LLDLF motif, 2.75 Å, blue) and Yap1801 (1.95 Å, grey). All peptides bind to three binding boxes: Clathrin, Arrestin and W boxes. Zoom: individual Chc-NTD boxes with Ent2 CBM-containing peptide.

(Ent1, Ent2 and Sla1) and an auxilin-like protein involved in clathrin uncoating (Swa2). In addition, we identified several adaptor proteins directly involved in trafficking processes. These include: Ent5; the beta chain of AP1 (Apl2); Gga1 and Gga2, all associated with clathrin-mediated trafficking between the Trans-Golgi Network (TGN) and endosomes. The delta chain of AP3 (Apl5), which regulates transport from the Golgi to vacuoles was also detected (Fig. 1b). Related to Golgi-endosome trafficking we also found proteins such as the Ypt/Rab transport GTPases activator, Mdr1, involved in vesicle protein trafficking; Inp5, a protein from the inositol polyphosphate 5-phosphatases family involved in vesicle trafficking and cell wall formation; Sbe22 and Bni4, related to the localization of Chs3 (Chitin synthase) between the Golgi and the membrane.

Other proteins containing a CBM but not involved in endocytosis or trafficking are Gtt3, a glutathione S-transferase; Lam6, a sterol transporter between cellular compartments; Smc5 involved in DNA repair and maintaining chromatin integrity; Uls1, a DNA helicase which mediates the ubiquitination of SUMO conjugates; Rim15, a protein kinase involved in nutrient regulation under nutrient-limited conditions; Bop3 involved in resistance to methyl-mercury; Ubp2, a ubiquitin hydrolase, Kar5, a pheromone involved in nuclear fusion; and Fsh3, a serine hydrolase involved in oxidative stress and apoptosis. At the moment there is no functional, biochemical and /or structural evidence that would support their binding to clathrin (Supplementary Table 1).

In contrast to what was observed for murine AP180 and other human proteins containing multiple CBMs[38], there is only one unique clathrin binding motif found per protein within yeast adaptors (Fig. 1b).

The Arrestin Motif ([LI][LI]GXL) is found in Scd5, a suppressor of clathrin deficiency related to actin filament network regulation[39], or in TGN trafficking (Supplementary Table 1). The W-box motif (WxxW) is

not present in the yeast proteome. However, similar to their human counterpart, many yeast CBMs seem to contain a negative charge in the +5 or +6 positions (Fig. 1b).

## ScCHC-NTD complexes reveal occupancy of three Chc boxes

To investigate the existence of binding sites present in Clathrin NTD, we crystallized the N-terminal domain of Chc with different peptides derived from adaptor proteins containing CBMs. Here, we present the NTD complexes with Ent1, Ent2, Ent5.1 (LIDL peptide), Apl2.1 (LLDLF peptide) and Yap1801 (Fig. 2 and Supplementary Fig 1.). The NTD structures for all complexes are very similar with Cα RMSDs between 0.36 Å and 0.64 Å (compared to Ent2 as reference between residues 5 and 336: Ent1 = 0.64 Å, Ent5 = 0.57 Å, YAP1801 = 0.36 Å and APL2 = 0.53 Å) with the exception of the structure in complex with Apl2 that shows the formation of a terminal helix (residues 337–363), transitioning from a helix-turn-helix configuration to an extended, straight helix (Supplementary Fig. 2). All our structures show that each peptide binds to three different binding pockets (Clathrin, Arrestin and W-box) (Figs. 2, 3 and Supplementary Fig. 3). An interesting structural strategy observed is that the CBM-containing peptides adopted different conformations to fit in each of the binding pockets, something that could only be acquired in the context of IDRs. While the Arrestin and Clathrin boxes-bound peptides adopt an extended conformation, the W-box displays a 3–10 helix fold to bind in the pocket. Interestingly, peptides bound to the yeast W-box adopt the same conformation that has been observed at the mammalian NTD homologue despite not having the canonical WxxW motif[18]. All three boxes share three common features: a glutamine residue, hydrophobic pockets, and positively charged residues that mediate interactions with the peptides. The glutamine residue (Q89 in the Clathrin box, Q155 in the W-box and Q195 in the Arrestin box) behaves as an anchoring residue forming a hydrogen bond with the peptide's backbone (Fig. 2). Our structural analysis

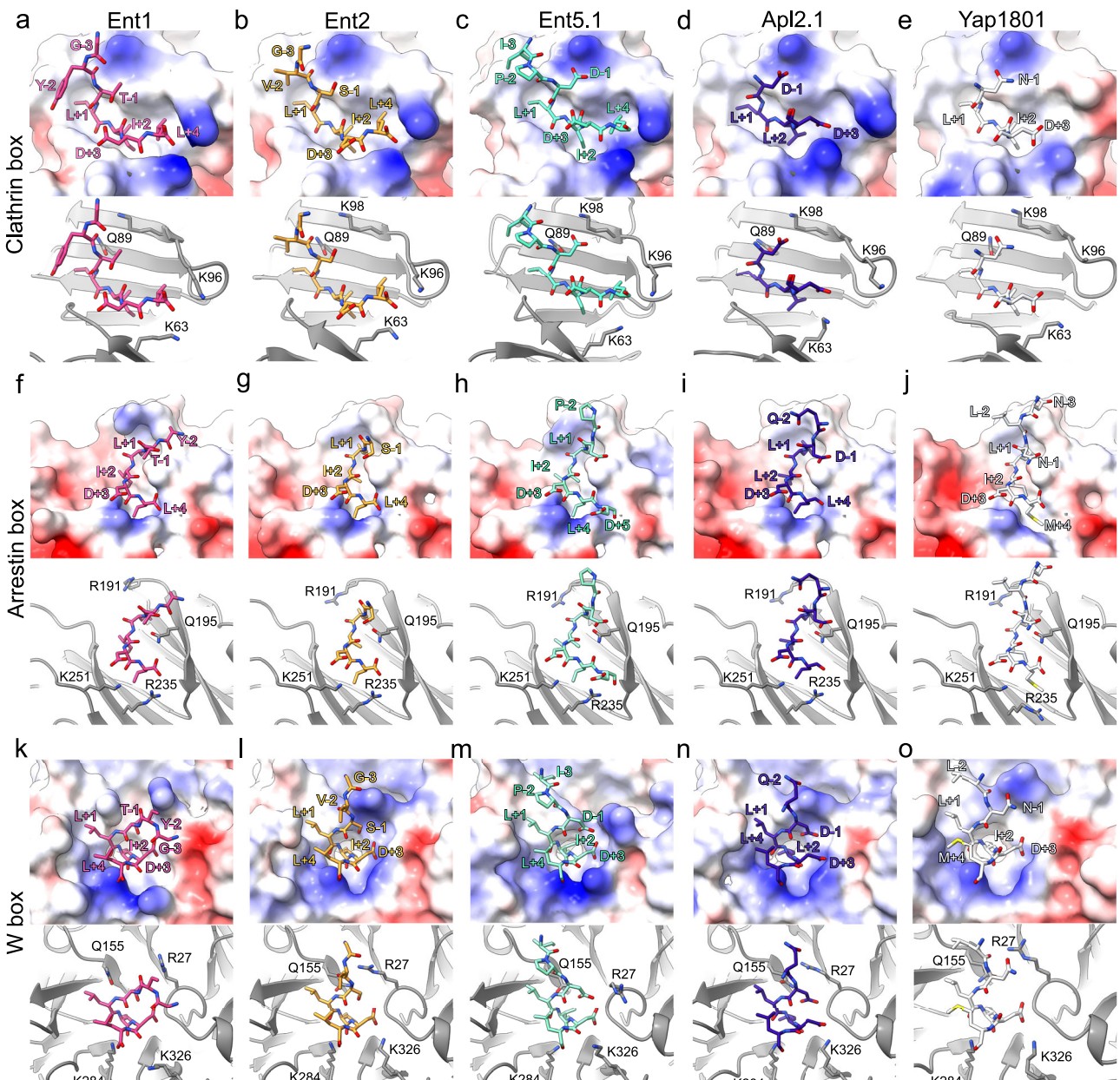

**Fig. 3 | Crystal structures of ScCHC NTD bound to CBM-containing peptides from Ent1 (carbon atoms in pink), Ent2 (orange), Ent5 (cyan), APL2 (blue), and YAP1801 (white). a** NTD-Ent1 complex in the Clathrin box shown in surface (top) and cartoon (bottom). **b** NTD-Ent2 complex in the Clathrin box shown in surface (top) and cartoon (bottom). **c** NTD-Ent5.1 complex in the Clathrin box shown in surface (top) and cartoon (bottom). **d** NTD-Apl2.1 complex in the Clathrin box shown in surface (top) and cartoon (bottom). **e** NTD-Yap1801 complex in the Clathrin box shown in surface (top) and cartoon (bottom). **f** NTD-Ent1 complex in the Arrestin box shown in surface (top) and cartoon (bottom). **g** NTD-Ent2 complex in the Arrestin box shown in surface (top) and cartoon (bottom). **h** NTD-Ent5.1 complex in the Arrestin box shown in surface (top) and cartoon (bottom). **i** NTD-Apl2.1 complex in the Arrestin box shown in surface (top) and cartoon (bottom). **j** NTD-Yap1801 complex in the Arrestin box shown in surface (top) and cartoon (bottom). **k** NTD-Ent1 complex in the W box shown in surface (top) and cartoon (bottom). **l** NTD-Ent2 complex in the W box shown in surface (top) and cartoon (bottom). **m** NTD-Ent5.1 complex in the W box shown in surface (top) and cartoon (bottom). **n** NTD-Apl2.1 complex in the W box shown in surface (top) and cartoon (bottom). **o** NTD-Yap1801 complex in the W box shown in surface (top) and cartoon (bottom). Surface electrostatics were calculated for the complexes. Peptides are shown in stick representation, and the NTD is shown in ribbons. Lysine (Lys) and Arginine (Arg) residues are labelled to indicate charged interactions between the peptide and NTD.

indicates that there are three equivalent Chc boxes for adaptors present in yeast NTD.

The three boxes showcase distinct hydrophobic pockets that accommodate hydrophobic residues at positions +1, +2, and +4 of the motif. In the Clathrin box, the hydrophobic pocket is formed by V49, V81, I87, F91, and L93. The Arrestin box features W167, L186, I197, F220, and I237. Meanwhile, the W-box exhibits a hydrophobic pocket formed by F26, I173, I180, F266, and I268 (see Figs. 2 and 3).

A network of salt bridges and hydrogen bond interactions exists between the peptide terminal carboxylate and Lys and Arg residues of each respective box. In the Clathrin box; K63 and K96 establish salt bridges with position +3 of the motif. While these interactions are consistently observed in all peptides, an exception is noted for Ent5.1, where no density for the Asp residues at positions +5 and +6 is observed (Fig. 3). Additionally, K98 interacts with position −2 of the motif as observed in the Ent5.1, Apl2.1 and Yap1801 complexes.

Backbone-backbone hydrogen interactions are observed for N64 and G66 with peptide residues at positions +1 and +3. For the Ent1 and Ent2 complexes, a water molecule bridges the interaction between the carbonyl oxygen at the motif's +3 position and N64 (Supplementary Fig. 4).

In the Arrestin box, salt bridges are observed between the peptide's Asp at position +3 and residues R235 and K251 from the NTD. A common feature to all complexes is a water-mediated interaction between the backbone carbonyl from E238 and the peptide's backbone on position +3 (Supplementary Fig. 3). Water-mediated interactions are also observed in the W-box between the NTD backbone (L24, D25 and V309) and the peptide at position +3, for both Ent1 and Ent2. However, Apl2.1 and Yap1801 do not show density compatible with water molecules near the binding sites. Residue K284 within the W-box establishes salt bridges with the carboxy-terminal of Ent1, Ent2 and Yap1801. Concurrently, K326 forms hydrogen bonds with the peptide backbone at positions +2, +4 and +1 for the Ent5.1, Apl2.1 and Yap1801 complexes.

When comparing the yeast and mammalian motifs (we refer to mammalian indistinctly in this section as all mammal clathrin NTD sequences of the first Chc isoform, including *Bos taurus* and *Homo sapiens*, share an identity of almost 100%, Supplementary Table 2) we observe that the mammalian motif can accept any residue at the position +3, while it is a conserved Asp in yeast. This residue forms a salt bridge with residue R235 from the yeast Arrestin box and is substituted by H229 in mammals (Supplementary Fig. 5a). In addition, N64 in yeast coordinates a water molecule in the Clathrin box and interacts with the CBM carbonyl atom at position +3 (Supplementary Fig. 3). This coordination is mediated by residue R64 in mammals, where the Arg sidechain forms a hydrogen bond with the +3 position of the mammalian motif (PDBID 5M5S, 7ZX4).

For the mammalian W-box, residues V177 and V262 allow the binding of WxxW peptides. The yeast W-box displays isoleucine at these positions. Interestingly, there are no WxxW motifs present in the *S. cerevisiae* proteome.

To expand our current understanding, we utilized AlphaFold (AF2 Multimer V2.3 and AF3)[40,41] to model other non-crystalized complexes of CBMs containing peptides with NTD. As a control, we compared our crystal structures with the predicted models (Supplementary Fig. 6 for AF2 and Supplementary Fig 7. for AF3 models). While for the Clathrin box, the predicted peptide position matches our experimental data, discrepancies are obtained for the Arrestin box, where AF2 failed to correctly place the Sla1 peptide (depicted within dashed red circles in Supplementary Fig 6). In addition, we observe differences between our crystal structures of the W-box and the predicted complexes for Ent1, Gga2, Swa2 and Sla1. In these predictions, the peptides are not accurately positioned, leading to clashes. However, the AF2 model successfully predicts binding between the W-box and a second CBM found in Ent5 (Ent5.2). The substitutions we have observed between *S. cerevisiae* and mammalian species appear to influence their binding capabilities for the W-box. While incapable of binding WxxW peptides, the yeast W-box is capable of accommodating CBMs with a LLDW sequence.

For AF3 models (see Supplementary Fig. 6), we observe an improvement for the Sla1 and Swa2 complexes predictions. Peptides bind in the W-box as observed in the crystal structures. However, issues with the predictions for Ent1 and Gga2 in the W-box persist. Additionally, while AF2 Multimer correctly predicted Ent2 and Yap1802 in the W-box, AF3 produced incorrect models for these peptides.

## Sc-CHC selectivity for adaptors revealed by binding affinities

In order to determine an apparent binding affinity of CBM containing peptides to the Chc-NTD, we performed NanoDSF (nDSF). A general observation is that NTD is thermo-stabilized in the presence of adaptor

peptides containing CBMs (Fig. 4a–f). The change in the melting temperature ($T_m$) for the bound and unbound states is greater than 15 °C. This increase in $T_m$ is explained by the formation of multiple favourable interactions, observed in our crystal structures. To determine apparent dissociation constants ($K_D^{App}$), we analysed the thermal shift produced by six different peptides: Ent1, Ent2, Ent5.1, Apl2.1, Yap1801, and Yap1802. Here, the underlying model assumes a one to one binding and therefore the $K_D^{App}$s represent global averaged values for all boxes.

Ent5.1 is characterized by an optimal binding motif, defined by the canonical LIDL[23] sequence flanked by Asp residues, and displays the lowest $K_D^{App}$ ~ 34 µM (Fig. 4c and g). The critical role played by the Asp on position −1 of the motif is exemplified in the structure of the Clathrin box, where Asp −1 forms a salt bridge with K98 (Fig. 3c).

Both Ent1 and Ent2 peptides share an identical LIDL[42] motif and display similar binding affinities ($K_D^{App} \cong 138$ µM and 177 µM, respectively). The lower affinity could be partially explained by a reduction in the absolute negative charges of the Ent1/2 and Ent5.1 CBMs (from −4 to −1). In addition, the substitution of Ile (from the LIDL motif) for Leu further reduces the binding affinity of Apl2.1 (residues 718-726) for NTD ($K_D^{App} \cong 245$ µM). Interestingly, the substitution of the second Leu (LIDL) and/or the addition of non-negatively charged residues at the C-terminal of the CBM decreases the binding affinity. Specifically, the Leu/Ile substitution on Yap1802 retrieves a $K_D^{App} \cong 311$ µM and the Leu/Met substitution on Yap1801 a $K_D^{App} \cong 598$ µM. (Fig. 4g). Overall, the fact that the measured $K_D$s for various CBMs are different, indicates that there is selectivity for adaptors at the NTD Chc.

To gain further insight into the different binding sites of Chc, we performed native mass spectrometry (native MS) experiments on Ent5.1 CBM, the strongest binder of the evaluated peptides (Supplementary Fig. 8). At experimentally suitable concentrations (up to 100 µM), we observed the appearance of Chc bound to three peptides and, although in much lower amounts, a fourth binding event was also detected (Supplementary Fig. 8a). The fourth binding event could either correspond to the Royle box or to unspecific binding. Interestingly, Chc displays an average occupation of two sites (Supplementary Fig. 8b). A qualitative inspection of the Scatchard plot (Supplementary Fig. 8c) reveals a strong deviation from a straight line, which is the expected result in the case of non-interacting binding sites. Therefore, Chc has (at least) two sets of binding sites with different binding affinities and/or Ent5.1 binds in a cooperative manner[43,44]. Indeed, the measured fractions differ from the expected ones for a non-interacting identical sites model (Supplementary Fig. 8d).

## Ent1 and Ent2 show distinctive binding to Chc boxes

To comprehend the interaction determinants between Ent1 and Ent2 with Clathrin in vivo we performed acceptor photobleaching FRET experiments between Ent1 or Ent2 endogenously tagged with mNeonGreen and Chc-mScarlet. The experiment was conducted in the presence of Latrunculin A, an inhibitor of actin polymerization that halts endocytosis, thus enabling the localization of endocytic pits and the measurement of protein-protein interactions. We engineered a series of NTD mutants based on our structural analysis, aiming to disrupt the binding between the CBMs and the Chc boxes (see Methods). The specific mutants designed are as follows: for the Clathrin box, we introduced mutations K63E, I87D, Q89A and K98E (referred as the Cla mutant); for the Arrestin box, Q195A, I197T and K251E (Arr mutant); and for the W-Box, we mutated residues F26A and Q155A (W mutant). A control demonstrating that donor fluorescence is not affected by the photobleaching process is shown in Supplementary Fig. 9 (see the Methods section for a description of the experimental setup).

The mutation of the Clathrin box showed a significant reduction in FRET efficiency for Ent1-mNeonGreen when compared to wildtype Chc (~8% for WT vs. ~5% for Cla) as shown in Fig. 5a, b, while neither

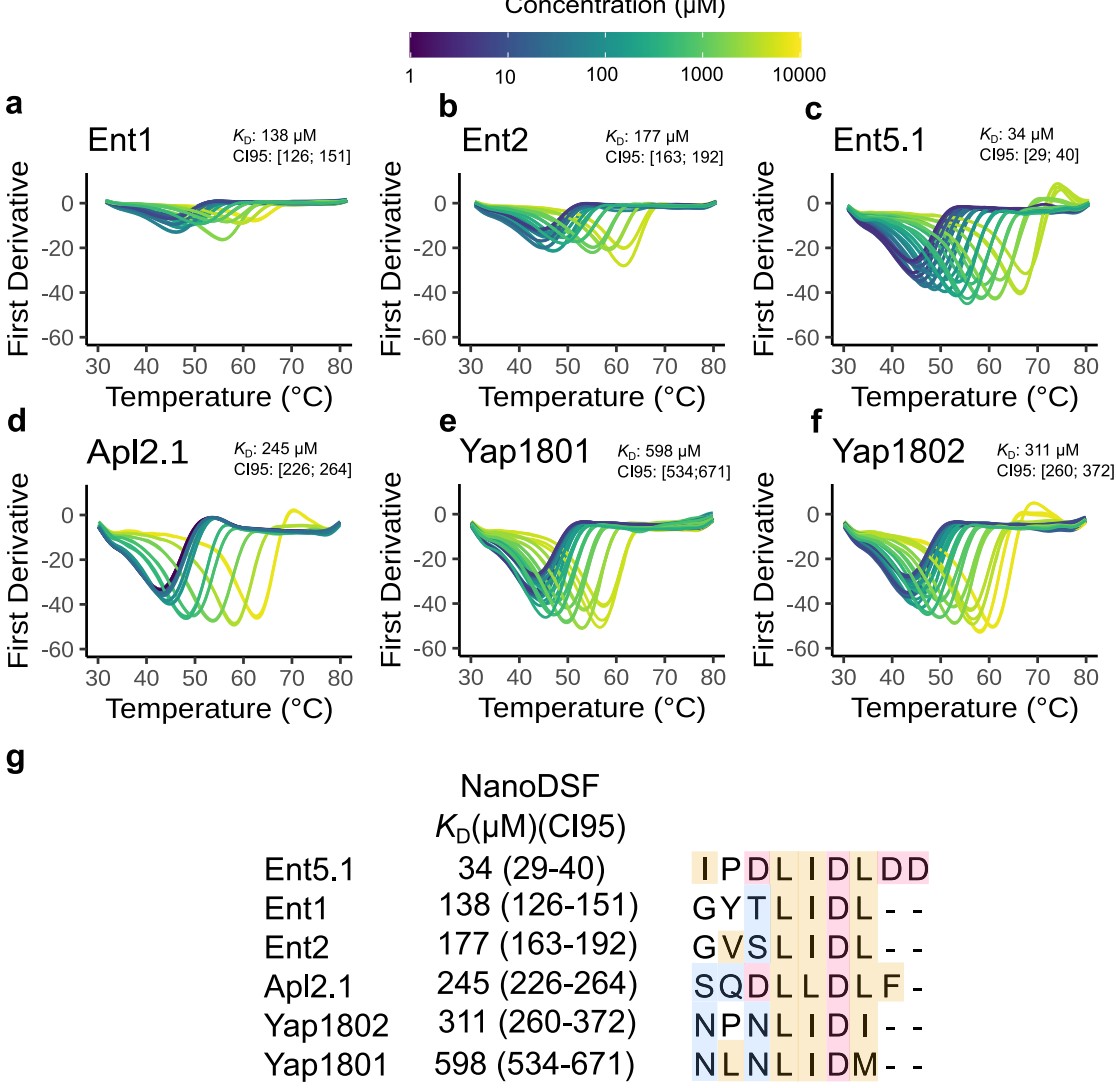

**Fig. 4 | Different APs show different binding affinities for NTD. a–f** Thermal denaturation first derivative curves followed at 350 nm display a thermal stabilization of the NTD upon binding to peptides containing CBMs observed by a shift of the $T_m$. **g** $K_D$ estimation for different CBMs using nDSF (see Methods). Colours in the sequence indicate type of amino acid: orange: hydrophobic; red: negatively charged; blue: polar. $T_m$: Melting temperature. CI95: Confidence Interval at 95%.

the Arr nor the W mutants exhibited any differences. Further examination of double mutants (Cla+Arr, Cla+W, and Arr+W), revealed that only Cla mutants exhibited a decrease in FRET efficiencies compared to WT (Cla+Arr: 5.25% and Cla+W: 5.93%). In addition, the triple mutant (Cla+Arr+W) showed the lowest FRET efficiency of 4.66% (Fig. 5b).

For the Chc-mScarlet/Ent2-mNeonGreen pair, mutation of the Arrestin box showed a significant decrease in FRET efficiency (9.02% for WT vs. 6.7% Arr), while the W and Cla mutants did not show significant changes. Among the double mutants, only Cla+Arr (6.1%) presented a difference when compared to WT. Similar to what has been observed for Ent1-mNeonGreen, the Cla+Arr+W mutant also exhibited a significant decrease in FRET efficiency compared to WT (Fig. 5c, d). These results indicate a preferential binding of Ent1 to the Clathrin box and of Ent2 to the Arrestin box.

Our in vitro experiments indicated that Ent1 is a stronger binder than Ent2, and the FRET experiments showed that they target preferentially the Clathrin and Arrestin boxes, respectively. To understand if Ent1 and Ent2 compete for the sites we have performed experiments at sub saturating concentrations to mimic initial recruiting events of both peptides and followed their binding to NTD by native MS (Supplementary Fig. 10 and Fig. 6). In the competitive landscape, Ent2

demonstrates reduced efficacy in countering Ent1's occupancy (Fig. 6b), corroborating what was observed in the FRET experiments. Conversely, Ent1 more effectively displaces Ent2 (Fig. 6a).

In order to understand if there is a preference between the sites for Ent1 or Ent2, we analysed the same NTD mutants previously described and estimated binding affinities in vitro using NanoDSF thermal shift (Supplementary Fig. 11). Mutation of the Clathrin box showed a reduction of almost seven-fold in the binding affinity to Ent1 ($K_D^{App}$ WT ≅ 138 μM vs. $K_D^{App}$ Cla ≅ 968 μM), while Ent2 exhibits a three-fold increase in its dissociation constant ($K_D^{App}$ Cla ≅ 509 μM). This data confirms Ent1's binding preference towards the Clathrin box. Mutations introduced in the Arrestin box have a comparable effect on the binding for both Ent1 ($K_D^{App}$ ≅ 166 μM) and Ent2 ($K_D^{App}$ ≅ 216 μM) peptides. In addition, W-box mutants also show comparable affinities for both epsins. The introduction of Cla double box mutations results in larger changes in binding affinities: Cla+Arr retrieved a $K_D^{App}$ Ent1 ≅ 2050 μM (15 fold increase), while for Ent2 the decrease in its affinities is around 5 fold. The Cla+W mutants have a deficient binding of almost nine-fold for both Ent1 and Ent2. However, the Arr+W mutant shows similar $K_D^{App}$s to that obtained for the WT NTD confirming that the Clathrin box is dominating. Finally, when mutating the three boxes (Cla

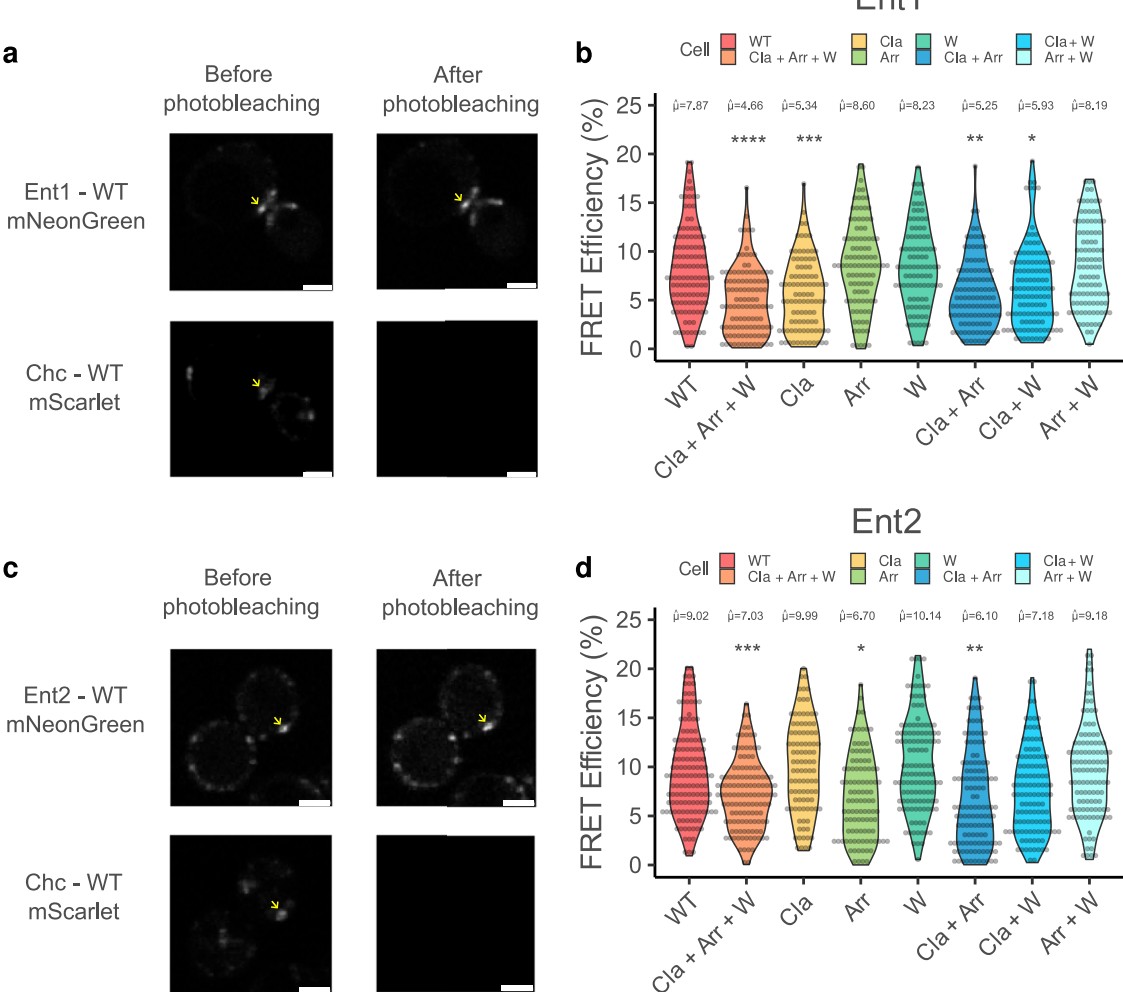

**Fig. 5 | In vivo FRET experiments show a preference for Ent1 for the Clathrin box and for Ent2 for the Arrestin box. a** Representative images (from four different microscopy sessions) from the FRET acceptor photobleaching experiment showing Ent1-mNeonGreen and Chc-mScarlet, both before and after the photobleaching of mScarlet. An arrow marks a location where FRET calculations were performed. Scale bars, 5 μm. **b** FRET efficiency distribution (%) for each pair, consisting of Ent1-mNeonGreen and various Chc mutants (WT: red, Cla+Arr+W: orange, Cla: yellow, Arr: green, W: turquoise, Cla+Arr: blue, Cla+W: light blue, Arr+W: cyan). **c** Representative images (from four different microscopy sessions) from the Ent2-Chc FRET Acceptor photobleaching experiment featuring Ent2-mNeonGreen and Chc-mScarlet. Scale bars, 5 μm. **d** FRET efficiency (%) distribution for each pair, consisting of Ent2-WT-mNeonGreen and various Chc mutants (WT: red, Cla+Arr+W:

orange, Cla: yellow, Arr: green, W: turquoise, Cla+Arr: blue, Cla+W: light blue, Arr +W: cyan). **b,d** The data encompasses a minimum of 98 stalled endocytic events per FRET pair. The numbers above each plot represent the trimmed mean value for the respective FRET pair. Asterisks indicate statistically significant differences (Ent1 $p$-value: ****Cla-Arr-W = 3.41e$^{-6}$, ***Cla=1.43e$^{-3}$, **Cla-Arr=1.64e$^{-4}$, *Cla+W = 0.02, Ent2 $p$-value: ***Cla-Arr-W: 6.4e$^{-6}$, *Arr: 5.85e$^{-5}$, **Cla-Arr:8.15e$^{-7}$), using two-sided Yuen trimmed means, corrected for multiple comparisons by Holm-Bonferroni method) compared to WT. WT: Wild-type. Cla: Clathrin box Mutant, Arr: Arrestin box mutant, W: W box mutant, Cla + Arr: Clathrin and Arrestin box mutant, Cla + W: Clathrin and W box mutants, Arr+W: Arrestin and B box mutant, Cla+Arr+W: Clathrin, Arrestin and W box mutant.

+Arr+W), the binding affinity reduction was the strongest for Ent1 ($K_D^{App} \cong 3450$ μM, almost 25x) while the $K_D^{App}$ for Ent2 is $\cong 1040$ μM.

**Clathrin Box mutations decrease Ent1-Abp1 endocytic events**
To evaluate the impact of mutations in the Clathrin boxes during endocytosis, we measured lifetime of Ent1 and Ent2 proteins endogenously tagged with mNeonGreen in the presence of Abp1-mTurquoise2 at endocytic sites by TIRF microscopy. Abp1 is involved in the recruitment of Actin (necessary for membrane invagination and vesicle formation) and therefore is associated with a productive endocytic event while its absence towards the end of Ent1/Ent2 lifetime would represent an abortive site. We have classified endocytic events into two main categories (using the programme cmeAnalysis[45]): Ent1/Ent2 and Abp1 positive (Abp1 + ) or Ent1/Ent2 and Abp1 negative (Abp1-). First, we have examined the behaviour of Ent1 and Ent2 at Abp1+ sites and observed that they follow distinct endocytic

recruitment dynamics (Fig. 7). Ent1 is progressively recruited to the endocytic pit until the appearance of Abp1 (Fig. 7a, c). In contrast, Ent2 recruitment remains almost constant throughout the event (Fig. 7b, c). Next, we compared the frequency of Ent1 or Ent2 at endocytic sites which are Abp + , between WT and Cla+Arr double mutant cells. This mutant has shown a lower binding in vitro affinity for both adaptors (Supplementary Fig. 11). The distribution plot of endocytic events for Ent1 shows a decrease in the percentage of Abp1+ sites (WT~33% vs. Cla+Arr ~15%). This is supported by decreased Abp1 presence in combined NTD Cla mutants (Supplementary Fig. 12). For Ent2, the total number of Abp1+ sites is lower than those observed for Ent1 indicating its less frequent presence (Fig. 7d).

## Discussion
We have identified a yeast Chc binding motif defined as [DSNTV]L[IL] D[ILMFW], which shares similarities with its mammalian counterpart

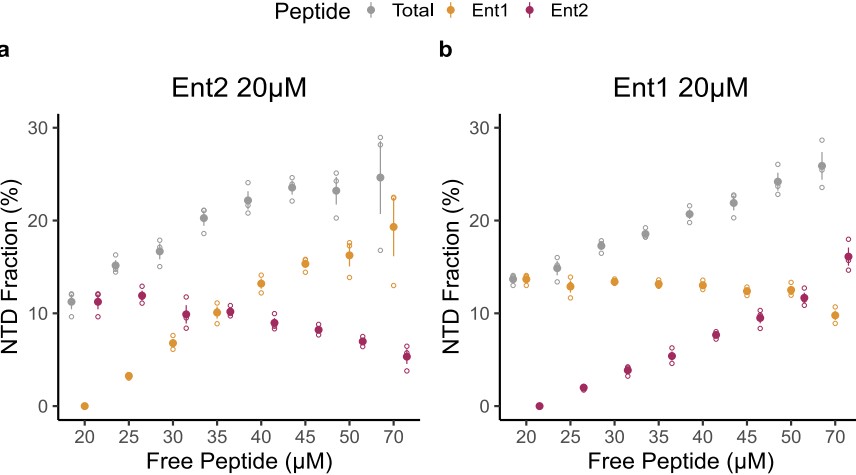

**Fig. 6 | One-site-occupancy binding analysis of Ent1 and Ent2 CBMs to Chc-NTD.**
**a** Chc-NTD bound fraction in a titration of Ent1 peptide ranging from 0 to 50 μM in the presence of 20 μM Ent2 peptide (20 to 70 μM total peptide concentration) ($n = 3$). **b** Chc-NTD bound fraction in a titration of Ent2 peptide ranging from 0 to 50 μM in the presence of 20 μM Ent1 peptide (20 to 70 μM total peptide concentration) ($n = 3$). Filled points represent the mean fraction of total Chc-NTD bound to the peptide, while empty points represent individual measurements. Error bars indicate the standard error (SE) of the mean.

(LΦXxΦ[D/E], where Φ is any hydrophobic residue). The majority of proteins containing this motif are recognized as clathrin-interacting partners, participating predominantly in processes such as endocytosis or trafficking between the trans-Golgi network (TGN) and other cellular compartments. The discovered CBM does not require a negative charge at position +5, described to be essential in mammals, but the presence of an aspartic acid at position +3, which mediates interactions with the positively charged residues at the NTD binding sites. The CBM showing the highest affinity for NTD, belongs to Ent5 and is characterized by the presence of negative charges surrounding the core motif. Unlike what has been observed in mammals, yeast adaptor proteins typically lack multiple CBMs within their sequences[38]. It's worth noting that yeast cells lacking clathrin remain viable. Significantly, key trafficking proteins such as Gga1, Apl2, and Ent5 contain two motifs, indicating clathrin interaction might be more relevant during sorting. Additionally, our study highlights the absence of specialized motifs within the yeast proteome that are prevalent in higher eukaryotes, such as the WxxW motif[18]. These differences help in understanding fundamental differences in the molecular mechanisms of protein recruitment and endocytosis when comparing yeast to higher eukaryotes.

A distinct subset of proteins containing the CBM is implicated in ubiquitin-mediated processes. Studies have demonstrated that ubiquitination of Ede1 is crucial for endocytosis[46]. Additionally, Eps15, the human homologue of Ede1, undergoes polyubiquitination[47]. The presence of a CBM in Ubp2, a Ubiquitin hydrolase, could recruit this protein at subsequent stages, suggesting a ubiquitin regulatory mechanism within the endocytic process.

Interestingly, the majority of proteins containing the proposed CBM were not identified as clathrin binders in the comprehensive proteome interactome study conducted by Michaelis et al. in 2023[48]. Only Ent2's clathrin interaction could be detected among all the well-established clathrin binders such as Ent1, Yap1801, Yap1802, or Sla1, which are absent in the mentioned study. Additionally, proteins associated with trafficking at the TGN, where Clathrin is actively involved, were also not identified in this screening. The interactome study is based on forward and reverse pull-downs that may underrepresent micromolar interactions. Similar observations can be found in the large-scale analyses by Benz et al[49]. focused on the human proteome. Utilizing phage display technology, they identified some clathrin binders featuring the LIx[FW] motif, however missing the complexity of interaction networks. This highlights the challenge of capturing transient interactions in large-scale studies.

The NTD-CBM complexes we examined in this study share several key characteristics with their mammalian equivalents, including a hydrophobic pocket and positively charged residues. These residues primarily interact with the C-terminus of the CBMs, while water molecules facilitate backbone-backbone interactions. Water-mediated interactions, particularly those found in the Clathrin and Arrestin boxes, are crucial for stabilizing the peptide's backbone and facilitating its binding to NTD. These structural waters have also been observed in the *Bos taurus* Chc-NTD crystal structures[22]. Water molecules may provide versatility to the scaffold, adaptable to the dynamic requirements presented by different adaptors at the binding site, suggesting an evolutionary preference for transient secondary structures over sequence specificity.

Our crystal structures show that one peptide binds to three distinct binding sites: Clathrin, Arrestin, and W-boxes. This triple binding has also been demonstrated in NMR experiments on the mammalian system[21]. A unique feature observed in the yeast system is that the NTD residues forming the W-box would not support the binding of a peptide with a WxxW motif, which is actually not found within the yeast proteome. Instead, the W-box accommodates a Trp at CBM position +4, as shown for one of the Ent5.2 CBMs.

An unexpected structural difference observed for the crystal complex with Apl2.1 is the formation of a terminal helix, enabled by the dimerization of two NTD molecules. The presence of alternative secondary structure elements could introduce differences in the overall stability of the Chc complexes or could suggest a mechanism for transmitting signals from the N-terminal domain to the Chc repeat region. Whether these dynamic structural features could play a crucial role in modulating clathrin polymerization or a transition in the curvature of the lattice would need to be investigated. So far, it has been demonstrated that lattice assembly is tightly regulated by the ratio of clathrin to adaptor proteins[50,51].

Given that several adaptors could compete for Chc binding sites, at similar endogenous concentrations, occupancy could be driven in a 'first come, first served' manner. However, our biophysical experiments indicate differences in dissociation constants across various CBM-containing peptides, with more than an order of magnitude separating the strongest and weakest binders. This suggests a competitive scenario and, therefore, hierarchical binding in clathrin-mediated processes. The binding of adaptor peptides to the NTD increases its stability, with nDSF thermal shifts of over 15 °C. This enhanced stability of the NTD-CBM complexes has likely resulted in

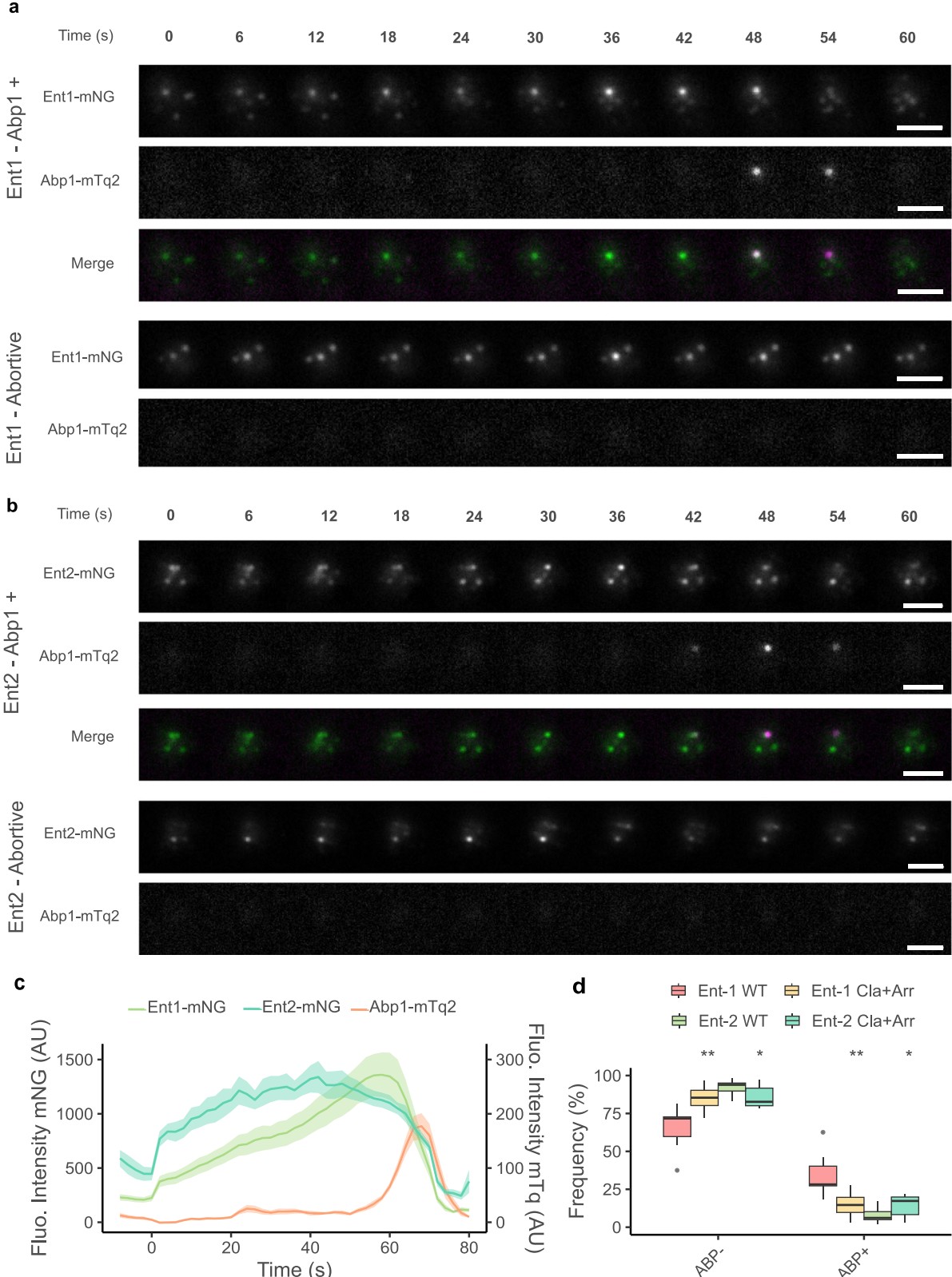

evolutionary advantages for the clathrin scaffold. The observed low affinity of these interactions introduces a dynamic aspect to the network, allowing interactions to form and dissociate following local concentration changes and leading to a more stable structure that can be tightly regulated. Mutagenesis experiments conducted on clathrin binding sites revealed a consistent preference of Ent1 for the Clathrin box, with the Arrestin and W-boxes appearing to play a less critical role

for Ent1, serving as secondary sites of interaction. Ent2's main target site appears to be the Arrestin box as shown in our FRET in vivo experiments. Competition experiments conducted at sub saturating concentrations using native MS revealed that Ent1 is able to displace Ent2 but not vice versa.

When mutating both the Cla and Arr boxes and tracking Ent1 at endocytic sites in vivo, we observe a significant reduction in Abp1

**Fig. 7 | Mutations on the Clathrin and Arrestin boxes increase the number of abortive Ent1-mediated endocytic events. a,b** Timelapse of a single endocytic event showcasing co-localization of Ent1-mNeonGreen (Green) (**a**) or Ent2-mNeonGreen (Green) (**b**) and Abp1-mTurquoise2 (Magenta) in a CHC wild-type cell. For each experiment, an example of AbpP+ and an abortive event are shown. Scale bar, representing 2 μm. **c** Average fluorescence intensity traces for Ent1-mNeonGreen (Green) or Ent2-mNeonGreen (Cyan) and Abp1-mTurquoise2 in Ent1-tagged cells (orange). Intensity data are presented as mean ± standard error (SE). **d** Frequency boxplot of Ent-1-mNeonGreen and Ent-2-mNeonGreen Abp+ and Abp- events in WT (Ent1 n = 15 and Ent2 n = 18) and Cla + Arr mutants (Ent1 n = 14 and Ent2 n = 13). For each condition, the central line represents the median, while the box represents the first quartile (Q1) and third quartile (Q3). The whiskers extend to the smallest and largest data points that are within 1.5 times the interquartile range

(IQR) from the quartiles. Ent1 WT ABP-: minima = 54, Q1 = 59.75, median = 71.7, Q3 = 72.75, maxima = 81.4. Ent1 Cla+Arr ABP-: minima = 72.2, Q1 = 80.225, median = 85.4, Q3 = 90.275, maxima = 96.7. Ent2 WT ABP-: minima = 83.1, Q1 = 89.8, median = 93.9, Q3 = 95.05, maxima = 98.1. Ent2 Cla+Arr ABP-: minima = 78.2, Q1 = 80.1, median = 82.7, Q3 = 91.7, maxima = 97. Ent1 WT ABP + : minima = 18.5, Q1 = 27.25, median = 28.2, Q3 = 40.25, maxima = 46. Ent1 Cla+Arr ABP + : minima = 3.3, Q1 = 9.725, median = 14.65, Q3 = 19.675, maxima = 27.8. Ent2 WT ABP + : minima = 1.9, Q1 = 4.95, median = 6.1, Q3 = 10.25, maxima = 16.9. Ent2 Cla+Arr ABP + : minima = 3, Q1 = 8.3, median = 17.3, Q3 = 19.9, maxima = 21.8. Asterisks indicates a singnificant difference in a two-sided Mann–Whitney U-test corrected for multiple comparisons (p-value: ** Ent1 = 2.3e$^{-5}$, *Ent2 = 0.015*), denoting statistical significance in a two sided Mann–Whitney U-test corrected for multiple comparisons using Benjamini and Hochberg.

positive (Abp1 + ) hence productive endocytic events. When examining the inclusion of Ent2 in the context of this Clathrin double mutant, we observe an increased inclusion in Abp1+ events, attributable to diminished binding from Ent1, which has now lost its main binding site.

Our work provides molecular insights into a potential hierarchy and selectivity of adaptor proteins for clathrin binding. It shows that while these proteins can compete for binding sites, they have also evolved to target different sites, making it likely that a single NTD can be occupied by different adaptors. Ent1 and Ent2 behave as different binders, and from our results, it can be speculated that Ent1 forms a more stable complex and likely plays a pivotal role in connecting the actin cytoskeleton to the membrane. Ent2 interactions are weaker, especially in the presence of Ent1 as a competitive binding partner. It has been previously observed that despite the analogous Ent1 and Ent2 containing conserved actin-binding domains, only Ent1 can bind the actin cytoskeleton productively at the endocytic site[30]. Here, we show that Ent1 binds more efficiently to Chc and therefore can be better recruited than Ent2. The hypothesis, based on a single pulldown experiment, which postulated in 2003 that Ent2 bound clathrin more efficiently than Ent1[42], can now be discarded. Additionally, the observation that the Ent1 actin-binding domain is more efficient in connecting the actin cytoskeleton to the endocytic coat can be re-interpreted since we now understand that Ent1 is actually more efficiently recruited by clathrin.

The potential simultaneous binding of an adaptor to three sites would be an efficient mechanism for adaptor recruitment. The presence of multiple binding sites allows for specialization, as observed in our FRET experiments. Specifically, we observe a preference of Ent1 for the Clathrin box and of Ent2 for the Arrestin box. Our TIRF experiments provide further evidence supporting this model. When both the Cla and Arr binding sites were mutated, we observed a significant reduction in productive endocytic events.

## Methods

### Protein expression and purification
The *Saccharomyces cerevisiae* clathrin heavy chain N-terminal domain (ScChc-NTD, 1–369) was cloned in a pETM-30 vector which contains a His-GST tag and a TEV cleavage site. Briefly, chemo-competent *E. coli* BL21 (DE3) cells were transformed with 100 ng of plasmidic DNA and grown overnight at 37 °C with 30 μg/ml of kanamycin. For protein expression, a 1:100 dilution of the preculture was done in 2xTY (16 g tryptone, 10 g yeast extract, 5 g NaCl per liter). Cells were grown at 37 °C until reaching OD 0.8–1. Then, the temperature was reduced to 18 °C, and induction was achieved by adding IPTG to a final concentration of 0.3 mM. Induction was performed overnight. Cells were harvested at 12,000 x g for 15 min at 18 °C and stored at −20 °C until purification.

Cells were resuspended in 5 to 10 mL of lysis buffer (20 mM sodium phosphate pH 7.5, 200 mM NaCl, 0.05% (w/v) Tween-20, 2 mM $MgCl_2$, 0.5 mM TCEP, +400 U DNAse I, 12.5 mM imidazole and a tablet of Complete EDTA-free protease inhibitor cocktail, Roche, per 100 mL

of buffer) per gram of cells. Rupture of the cells was achieved by using an Emulsiflex C3 (Avestin, Ottawa, Canada) cell disruptor at 15 kPsi three times. A centrifugation at 42,000 × g for 1 h at 4 °C was performed to clear the lysate.

The lysate was filtered with a 0.45 μm filter and loaded onto a Ni-NTA (Carl Roth, Germany) gravity column equilibrated with buffer A (20 mM sodium phosphate pH 7.5, 500 mM NaCl and 12.5 mM imidazole). The column was washed with 10 column volumes (CV) of buffer A and eluted with buffer B (20 mM sodium phosphate pH 7.5, 500 mM NaCl and 250 mM imidazole), collecting 1 mL fractions. The purity of the protein samples was assessed using SDS-PAGE. Subsequent to this analysis, selected fractions were pooled and treated with TEV protease at a ratio of 1 mg of TEV per 25 mg of protein. This mixture was then diluted in a 1:1 ratio with a dialysis buffer composed of 20 mM TRIS (pH 7.5), 200 mM NaCl, and 0.5 mM TCEP. Finally, the solution was subjected to overnight dialysis at 4 °C using the same buffer.

A reverse Ni-NTA was performed to remove HisTag-TEV and the cleaved HisGST from the solution. A prepacked Ni-NTA column was equilibrated with dialysis buffer and the flowthrough was collected and concentrated to 2 mL using a 10 kDa MWCO concentrator (Amicon Ultra 15, Millipore). The sample was loaded onto a Superdex 200 HiLoad 16/600 size-exclusion chromatography (SEC) column equilibrated with SEC buffer (50 mM Tris pH 9, 150 mM NaCl, 0.5 mM TCEP). Fractions were analysed for purity using SDS-PAGE, pooled, concentrated to 20 mg/ml, flash frozen in liquid nitrogen, and stored at -80 °C until used.

### Peptide synthesis
Peptides were purchased from NovoProLabs (Shanghai, China) with a purity of at least 98% and TFA-free (Acetate-salt). Ent1 (GYTLIDL), Ent2 (GVSLIDL), Ent5.1 (IPDLIDLDD), Yap1801 (NLNLIDM), Yap1802 (NPNLIDI) and Apl2.1 (SQDLLDLF). Unless stated otherwise, they were solubilized in SEC buffer at 10–20 mM concentrations. Each peptide was weighted in an analytical scale and solubilized with SEC buffer using their respective molecular masses. Concentration was corroborated with a refractometer for a selected number of peptides.

### Protein crystallization and data collection
All the protein crystallization experiments were performed in 3-lens Crystallization Swissci plates (SWISSCI, Zug, Switzerland) by sitting drop evaporation. NTD was incubated with excess peptide (between 23 and 40 fold) for 15 min at room temperature (21 °C) prior to setting-up the crystal plates. Ent1 CBM/Chc-NTD complex was crystallized with a 2:1 (protein:precipitant) ratio in a 0.2 M potassium/sodium tartrate and 20 %(w/v) PEG 3350 solution. The protein concentration was 10.4 mg/ml, while the peptide concentration was 10 mM (40:1 peptide:protein molar ratio). The first crystals appeared after one day and were harvested after 7 days. Ent2 complex was crystallized with a 2:1 (protein:precipitant) ratio in a 0.2 M lithium sulphate, 0.1 M Bis-Tris pH 5.5, 25%(w/v) PEG 3350 solution at 19°C. The protein concentration was 10.4 mg/ml, while the peptide concentration was 8 mM (30:1

peptide:protein molar ratio). The first crystals appeared after 3 days and were harvested after 7 days. Ent5.1 complex was crystallized with a 1:1 (protein:precipitant) ratio in a 10 % (w/v) PEG 1000, 10% (w/v) PEG 8000 solution at 19 °C. The protein concentration was 9 mg/ml, while the peptide concentration was 5 mM (23:1 peptide:protein molar ratio). The first crystals appeared after 3 days and were harvested 84 days after setting up the plate. Apl2.1 complex was crystallized with a 1:2 ratio (protein:precipitant) in a 4 M sodium formate solution at 19°C. The protein concentration was 9 mg/ml, while the peptide concentration was 5 mM (23:1 peptide:protein molar ratio). Crystals appeared after 84 days and were harvested the following day. Yap1801 complex was crystallized with a 2:1 ratio (protein:precipitant) in a 0.1 M Tris-HCl pH 8.5, 0.28 M lithium sulphate monohydrate, 30% (w/v) PEG 4000 solution at 4 °C. The protein concentration was 10 mg/ml, while the peptide concentration was 5 mM (20:1 peptide:protein molar ratio). Crystals appeared after 2 days and were harvested after 10 days. All complexes were cryoprotected using the respective mother liquor solution supplemented by 5% glycerol and peptide at the corresponding concentration.

Ent1, Ent2, Ent5.1, Apl2.1, and Yap1801 datasets were collected at P14 or P13 operated by EMBL at the PETRA III storage ring (DESY, Hamburg, Germany) (see Supplementary Table 3).

### Crystallography data processing

Data sets were merged using XDS (5.8.0425) and scaled using AIMLESS(0.7.4)[52]. The ScChc-NTD/peptide complex structures were solved by the molecular replacement method, using MOLREP (11.9.02)[53] and using as a search model an ScChc-NTD Model (1-369) made with PHYRE2[54] for ScChc-NTD/Yap1801 complex. Then, this refined model was used as a search model for the following structures. Iterative refinement and model-building cycles were performed using REFMAC(5.8.0425)[55] and Coot[56] from the CPP4i2 suite of programmes (v4.8)[57]. Regarding the refinement for Chc-Apl2.1 – we have included secondary structure restraints, NCS torsion restrains, TLS refinement (at the beginning of the refinement one per chain and after the initial cycles the different TLS groups were calculated using phenix), grouped Bfactor refinement using PHENIX (1.21_5207)[58,59]. The final steps of model building were also assisted with ISOLDE(1.0b3)[60] to correct the models for clashes. Data collection and refinement statistics are shown in Supplementary Table 3. Coordinates have been deposited to the PDB with accession codes 9EXG, 9EX5, 9EXF, 9EXT and 9EYT.

### Mutations in pETM-30-ScCHC-NTD and pRS315-5′UTR CHC-mScarlet

Mutagenesis was performed according to the following criteria based on the chemical property of the aminoacid: Lys/Arg residues are substituted with residues of opposite charge Asp/Glu. Polar residues are replaced by alanines and hydrophobic residues by Ala/Ser/Thr. In addition, we have selected combinations of mutants that could affect the interaction with targets but not compromise the thermal stability of the CHC-NTD fold, as predicted by FoldX 5[61] using a threshold of 1 kcal/mol.

For generating point mutations in both plasmids, we used a modified version of the quikchange protocol. Overlapping primers were designed (Supplementary Table 4) to copy the whole plasmid. The reaction mixture had 100 ng of template DNA, 200 nM of forward and reverse primers, 2x of Phusion mixture (40 mM TRIS pH 8.8, 4 mM MgCl$_2$, 120 mM KCl, 20 mM (NH$_4$)$_2$SO$_4$, 0.02 mM EDTA, 0.2% TritonX-100, 8% glycerol, 0.005% Xylene Cyanol FF, 0.05% Orange G, 0.4 mM dNTPs, 0.04 U/µL Phu-Sso7d Polymerase) and nuclease-free water to a total volume of 50 µl. The PCR programme used consisted of a first denaturation step at 98 °C for 3 min followed by 20 cycles of: (a) 1 min at 98 °C, (b) 45 s at 56 °C, and (c) 7 min at 72 °C. A final extension of 10 min at 72 °C was performed. After the PCR reaction, 1 uL of DpnI restriction enzyme was added to the reaction, and the mix was incubated for 2 h at 37 °C. 10 uL of the reaction was used to transform *E. coli*

DH5α cells. Colonies were selected and grown overnight in LB to extract plasmid DNA using a MiniPrep kit (NEB, Massachusetts, USA) using manufacturer's protocol. The identity of each mutant was confirmed by Sanger sequencing (Microsynth, Göttingen, Germany).

### NanoDSF

ScChc-NTD (8 µM) was incubated with the different peptides for 15 minutes and subsequently heated from 20 °C to 95 °C at 1 °C/min using a Nanotemper Prometheus NT.48 fluorimeter (Nanotemper) controlled by the PR.ThermControl software (version 2.3.1). The starting concentrations were as follows: Ent1: 7.5 mM, Ent2: 5 mM, Ent5.1: 3.75 mM, Apl2.1: 7.5 mM, Yap1801: 5 mM and Yap1802: 7.5 mM. In all cases, eleven peptide concentrations were used (1:1 dilutions).

To monitor the interactions of the ScChc-NTD mutants with Apl2.1, Ent1 and Ent2, the protein concentration was 10 µM. Also, for Ent1 and Ent2, the thermal ramp went up to 80 °C.

To determine the apparent $K_D$s, we applied the $T_m$ shift model[62] in the FoldAffinity module of the eSPC platform (spc.embl-hamburg.de)[62,63]. The $T_m$s were calculated by using the maximum (or minimum) of the first derivative at 350 nm.

### Native mass spectrometry

ScChc-NTD stocks were doubly buffer-exchanged into a 200 mM NH$_4$AcO, pH 9.0 buffered solution by two subsequent runs on P6 biospins (BioRad); the protein concentration was determined photometrically ($\varepsilon_{280\ nm} = 26{,}930\ M^{-1}cm^{-1}$) and the protein was diluted into 20 µl aliquots. Final samples contained 1.75 µM protein. Samples for competition experiments were prepared similarly for constant (20 µM) Ent1 and varying (0, 5, 10, 15, 20, 25, 30 and 50 µM) Ent2 concentrations and vice versa. *Mass measurements.* Samples were placed into gold-coated capillaries and after ionization by nano-electrospraying mass-measured in a QE-UHMR orbitrap mass spectrometer (Thermo Fisher Scientific) as follows: 1. establishment of a ("wet") spray using a capillary voltage of 1.10-1.15 kV but with the syringe locked as to guarantee a relatively high backing pressure. 2. Once a stable spray was established, the capillary voltage was increased to 1.3-1.4 kV with simultaneous unlocking of the syringe to remove the backing pressure or, when necessary, venting the hose that connects the syringe and ion source. 3. re-decreasing capillary voltage to 1.10-1.15 kV. By doing so, success in the sense of as least as possible unspecific clusters could be easily monitored by spectrum appearance and distinct downward steps of the total ion current (TIC). Transfer capillary temperature was set at 50 °C, all other parameters being standard: detector and transfer setting at "low" and high, respectively; trapping gas pressure setting "7" typically yielding 1.5–1.6 mbar force and $2.6$–$2.7 \times 10^{-10}$ mbar ultrahigh vacuum; background/noise setting "4.64"; scan range $m/z$ 1000–10,000; resolving power 12,500; ISD 35 V and CE 50 V.

**Data evaluation.** Spectra were exported from the Xcalibur Qual Browser (Thermo, V4.1.31.9) into SigmaPlot (Inpixon, v15.0.0.13) and normalized.

### SLiM Discovery

SLiM Search 4[37] was used to search in the *Saccharomyces cerevisiae* proteome for proteins containing the motif iteratively until no further proteins related to endocytosis were found. A disorder cutoff of 0.5 was used to perform the search. Manual curation based on AlphaFold2 models deposited in UniProtKB was done for obtaining the final adaptor protein list, discarding proteins where the motif is in a structured region, only those regions with a low confidence and predicted as unstructured were considered (pLLTD<50).

### AF2.3 Multimer and AF3 Protein-peptide complex prediction

We used AF2.3 Multimer (Evans et al., 2022) with default parameters. In the case of AF3 (Abramson et al., 2024), the AlphaFoldServer was used

with one copy of Chc-NTD and three copies of each peptide for running the predictions. In all cases, the best model is shown in Supplementary Figs. 5 and 6. It is worth noting that when a poor prediction was detected, it was observed in all models.

### Generation of yeast cells expressing ScCHC-mScarlet and Abp1-mTurquoise2

Generation of competent yeast cells. A 5 mL preculture of cells was incubated overnight at 30 °C with shaking in the appropriate medium. The OD of the overnight culture was determined and a new 50 ml culture with fresh medium was inoculated to OD of 0.15. Cells were grown at 30 °C until reaching an $OD_{600nm}$ between 0.4 and 0.6. Cells were spun down at 3000 x $g$, at 4 °C for 5 min, discarding the supernatant. Cells were resuspended in 30 mL of sterile water and spun down again at 3000 x $g$, at 4 °C for 5 min, and the supernatant was discarded. Cells were resuspended in 1 ml of sterile water and spun down in a tabletop centrifuge at 4 °C for 5 min at 3000 x $g$, and the supernatant was discarded. Cells were resuspended in 300 μL competent cell solution (5% v/v glycerol; 10% v/v DMSO), aliquoted in 50 μL, and stored at −70 °C (cooled down as slowly as possible).

### Generation of chc1Δ cells expressing pRS315-scCHC_mScarlet

Competent *Saccharomyces cerevisiae* cells (MK100 WT: *MATa*; *his3Δ200*; *leu2-3,112*; *ura3-52*; *lys2-801*) with Ent1p and Ent2p endogenously tagged with mNeonGreen were transformed with a vector containing the 5' UTR of Clathrin Heavy Chain, the ORF of CHC1 fused in the 3' with mScarlet (pRS315-scCHC_mScarlet) containing the LEU auxotrophy factor cassette using the following protocol[64]. Cells were grown overnight in YPD media and fresh YPD media was inoculated to an $OD_{600nm}$ of 0.1–0.15. The culture was grown until an $OD_{600nm}$ of 0.4–0.6 was reached. Cells were spun down at 3000 g for 5 min, the supernatant was discarded, the pellet was resuspended in 30 mL cold $H_2O$ and spun down as before described, supernatant was discarded. Cells were resuspended with 1 mL of 100 mM lithium acetate and transferred to a 1.5 mL centrifuge tube, spun down for 15 s in a tabletop microcentrifuge. Supernatant was discarded and 400 μL of 100 mM lithium acetate was added to resuspend the cells. 50 μL of the cell suspension was mixed with precooled 240 μL 50% PEG 3350, 10 μL of plasmid *pRS315-scCHC_mScarlet* carrying CHC1 gene tagged with mScarlet and LEU selection marker and 25 μL of 2 mg/mL ssDNA was added and vortexed. Cells were incubated at 30 °C for 30 min and at 42 °C for 25 min. Transformation solution was spun down for 15 s, supernatant was discarded and 100 μL $H_2O$ were added to plate cells on SC-Leu Agar plates. The same protocol was followed for transformation to knock out endogenous CHC1, only the cells before transformation were grown in SC-Leu medium.

Once colonies were found, competent cells were prepared and transformed with a URA resistance cassette with homology arms to endogenous CHC1 gene (See Supplementary Table 5 for primer S2/S3 sequence) using the established PCR cassette protocol (pFA6a-KlURA3 Vector)[65]. These positive endogenous CHC-KO cells were again prepared as competent cells for a final round of transformation to tag Actin Binding Protein 1 (Abp1) with mTurquoise2 (pFA6a-mTurquoise2-hphNT1 plasmid).

### Transformation of yeast cells for ABP1-mTurquoise tagging

Previously generated competent cells were thawed quickly and spun down at 13000 x $g$ for 2 min at 4 °C, discarding the supernatant. Cells were resuspended in transformation solution by vortexing (260 μL 50% w/v PEG-3350, 36 μL 1 M lithium acetate, 50 μL 2 mg/mL ssDNA, 14 μL of DNA of interest). Cells were incubated at 42 °C for 40 minutes and immediately put on ice for 2 minutes. 3 mL of YPD was added to the transformation solution and was incubated for 6 h at 30 °C while shaking. Cells were sedimented, resuspended in sterile water, plated in the corresponding Agar plate for selection, and left between 2 and 5 days in an incubator at 30 °C. Colonies were picked and grown in 4 ml

of the respective medium at 30 °C overnight with shaking. Genomic DNA was extracted using the manufacturer's instructions using a YeaStar Genomic DNA kit (Catalog D2002, Zymo Research, Irvine, California, USA). Genomic DNA was subject to a PCR to check for the insertion of the corresponding tag or the CHC KO. For transformations with a hygromycin-resistance cassette, replica plating was performed using Wattman paper to transfer the colonies to a new plate before performing the genomic DNA extraction and PCR check.

### Acceptor photobleaching FRET

To determine the FRET Efficiency of the Ent1/2p-mNeonGreen/Chc-mScarlet pair, yeast cells previously described were grown overnight at 30 °C with shaking in a 24-well plate using LD(=low-fluorescence SD)-Trp⁻ Leu⁻ medium (Yeast Nitrogen base without aminoacids supplemented with the corresponding DropOut media, Formedium, CYN402). Cells were diluted in fresh medium with a starting $OD_{600nm}$ of 0.1 and allowed to grow at 30 °C with shaking for several hours (4–6 h) until they reached log phase ($OD_{600nm}$ 0.6–1.2). Micro slide 8-well glass bottom plates (Catalog 80807, Ibidi, Gräfelfing, Germany) were treated with 40 μL of a 1 mg/ml Concanavalin A (Catalog C2010, Sigma Aldrich, prepared in 10 mM Sodium phosphate buffer pH 6, 10 mM $CaCl_2$, 1 mM $MnCl_2$, 0.01% $NaN_3$,) solution, incubated for 5 min and then washed twice with 40 μL of fresh medium. 40 μL of cell suspension was applied, incubated for 5 min and removed. Then each well was washed once with 40 μL of fresh medium and once with a 20 μL of a Latrunculin A (Catalog BML-T119, Enzo Life Sciences, Farmingdale, NY, USA) solution (0.2 μL Latrunculin A 8.33 μg/μl in 20 μL of medium). Finally, 20 μL of a Latrunculin A solution was added and incubated for 10 minutes before starting the measurements[66]. As a FRET photobleaching control, Ent1-mNeonGreen-tagged cells were transformed with a vector expressing mScarlet under the CHC promoter (pRS315-mScarlet). These cells were treated with Latrunculin A to stall endocytosis, allowing accurate measurement of endocytic events (puncta) before and after photobleaching.

Measurements were done at room temperature (21 °C) with a spinning disk confocal Nikon Eclipse Ti2 equipped with 488 nm, 561 nm and 640 nm lasers and an iXon Ultra 888 EMCCD camera installed in the Advanced Light and Fluorescent Microscopy (ALFM) facility in CSSB Hamburg (DESY, Hamburg, Germany) using NIS Elements (v5.42.04). Three images with 488 nm and three with 561 nm illumination were taken for each field of view before photobleaching. The field of view was subjected to photobleaching by operating lasers 561 nm and 640 nm at 100 % power for 25 seconds. Finally, three images were taken after photobleaching with 488 nm and 561 nm illumination as before photobleaching. Depending on cell density, images were taken in 7 to 10 fields of view. Images shown in Fig. 5a and c are representative of four different microscopy sessions. Acceptor photobleaching FRET efficiency was determined using FIJI (v1.54 f)[67] and FRETcalcV5[68]. Pre and post-photobleaching images were stacked together, and the background was subtracted. To pick each endocytic patch, the Polygon tool was used to select the pixels around it. At least 90 different endocytic patches were used for each condition, with at least 85% acceptor bleaching (Ent1-mNeonGreen/Chc1-mScarlet and Ent2-mNeonGreen/Chc1-mScarlet). Sample size: Ent1: Ent1-WT ($n = 116$), Ent1-Cla ($n = 98$), Ent1-Arr ($n = 102$), Ent1-W ($n = 98$), Ent1 Clat+Arr ($n = 111$), Ent1-Clat+W ($n = 122$), Ent1-Arr+W ($n = 103$), Ent1-Cla+Arr+W ($n = 112$). In the case of Ent2, Ent2-WT ($n = 136$), Ent2-Cla ($n = 98$), Ent2-Arr ($n = 104$), Ent2-W ($n = 126$), Ent2-Cla+Arr ($n = 126$), Ent2-Cla+W ($n = 117$), Ent2-Arr+W ($n = 112$), Ent2-Cla+Arr+W ($n = 127$). Results were analysed with the ggstatplot[69] from R (4.2.3). Pairwise comparisons were done using Yuen's trimmed means and were corrected for multiple comparisons by Holm-Bonferroni method. (Ent1: $F_{trimmed-means} = (7218.95) = 13.39$, $p = 2.33e^{-14}$, $\xi = 0.37$, $CI_{95\%} = [0.31, 0.42]$, $n_{obs} = 862$; Ent2 $F_{trimmed-means} = (7238.13) = 10.85$, $p = 7.13e^{-12}$, $\xi = 0.32$, $CI_{95\%} = [0.26, 0.39]$, $n_{obs} = 946$)[70].

## Lifetime TIRF microscopy

To determine the effect of mutating the CHC-NTD Clathrin, Arrestin and W-boxes in how endocytosis progresses, cells tagged with Ent1/2p-mNeonGreen, Chc-mScarlet and Abp1-mTurquoise2 were used. Yeast cells previously described were grown overnight at 30 °C with shaking in a 24-well plate using LD(=low-fluorescence SD)-Trp⁻, Leu⁻ medium (yeast nitrogen base without amino acids supplemented with the corresponding DropOut media, Foredium, CYN402). Cells were diluted in fresh medium with a starting $OD_{600nm}$ of 0.1 and allowed to grow at 30 °C with shaking for several hours (4-6hs) until they reached log phase ($OD_{600nm}$ 0.6-1.2). Micro slide 8-well glass bottom plates (Catalog 80807, Ibidi, Gräfelfing, Germany) were treated with 50 μL of a 1 mg/ml concanavalin A (prepared in 10 mM sodium phosphate buffer pH 6, 10 mM $CaCl_2$, 1 mM $MnCl_2$, 0.01% $NaN_3$, Catalog C2010, Sigma Aldrich) solution, incubated for 5 min and then washed twice with 50 μL of fresh medium. 50 μL of cell suspension was applied, incubated for 5 min and removed. Then each well was washed twice with 50 μL of fresh medium. Finally, 50 μL of fresh medium was added.

TIRF Microscopy was done at room temperature (21 °C) using a Nikon Eclipse Ti2 microscope equipped with 405 nm and 488 nm lasers and an ORCA-Fusion BT Digital CMOS camera installed in the Advanced Light and Fluorescent Microscopy (ALFM) Facility in CSSB Hamburg (DESY, Hamburg, Germany) using NIS Elements (v5.42.04). An oil immersion 100x objective was used (NA 1.49). For each field of view, a 5-minute movie was taken. Exposure for each channel was 500 ms in a 2 s interval (0.5 fps for each channel). Depending on cell density, movies were taken in 7 to 10 fields of view. Background subtraction was done with FIJI[67]. cmeAnalysis[45] was used to track and classify the tracked particles. All Ent1/Ent2-positive events were categorized based on whether they could recruit Abp1 into the pit. Events were labelled as either Abp1+ (Abp1 recruited) or Abp1- (Abp1 not recruited). This classification was performed using the two-colour tracking functionality available in CMEAnalysis. For each system the following number of replicas were taken (Ent1-WT n = 15, Ent1-Cla+Arr n = 14, Ent2-WT n = 18, Ent2-Cla+Arr *n* = 13). Running parameters for tracking and classification were kept as default by the package. Data was plotted with ggplot2 and Pairwise comparisons were done using a Mann–Whitney U-test corrected for multiple comparisons using Benjamini and Hochberg as implemented in R (4.2.3).

## Large language models

ChatGPT (https://chat.openai.com/) was used as an aid to correct written text. The authors take full responsibility for the manuscript content and code.

## Reporting summary

Further information on research design is available in the Nature Portfolio Reporting Summary linked to this article.

## Data availability

All data supporting the findings of this study are available within the paper and its supplementary information. X-ray crystallographic data has been deposited in the Protein Data Bank with codes 9EXG, 9EX5, 9EXF, 9EXT and 9EYT. The native mass spectrometry proteomics data have been deposited to the ProteomeXchange Consortium via the PRIDE partner repository with the dataset identifier PXD052864. Source data are provided with this paper.

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

## Acknowledgements

We thank Prof. Dr. Henning Tidow for suggestions and manuscript proofreading. We acknowledge the staff of the EMBL P13/P14 beamline as well as the Sample Preparation and Characterisation (SPC) facility of EMBL Hamburg at PETRA III (DESY, Hamburg) for assistance (Stephan Niebling, Christian Günther, Angelica Struve and David Ruiz-Carrillo). LAD was partly funded by EMBL Interdisciplinary Postdoc Programme (EIPOD3) under Marie Curie Actions COFUND 664726. OB is partly funded by the EMBL Career Accelerator for Research Infrastructure

Scientists Fellowship Program (ARISE) under Marie Skłodowska-Curie Actions COFUND 945405. The Leibniz Institute for Virology (LIV) is supported by the Free and Hanseatic City of Hamburg and the Federal Ministry of Health (BMG). C.U. and K.K. were funded through EU Horizon 2020 ERC StG-2017 759661 grant. C.U. further acknowledges an equipment grant from the Free and Hanseatic City of Hamburg.

## Author contributions

L.A.D. produced the proteins for Nano Differential Scanning Fluorimetry (nDSF), Crystallography and Native mass spectrometry (native MS). L.A.D performed biophysical and structural experiments, and interpreted crystallography data with the help of I.B. Native MS experiments were performed by K.K. under the supervision of C.U. O.B. helped with the analysis of biophysical data (nDSF and Native MS). K.V. performed in vivo mutation and tagging in *S. cerevisiae* cells. L.A.D performed Fluorescence Resonance Energy Transfer (FRET) and Total Internal Reflection Fluorescence (TIRF) experiments, supervised by M.S. and R.T. T.K. designed figures. M.G.A conceived and supervised the project. L.A.D. and M.G.A. wrote the manuscript with input from all authors.

## Funding

## Competing interests

The authors declare no competing interests.
