## [Transparent Peer Review file · Nature Communications]

Subtleties in Clathrin Heavy Chain Binding Boxes provide selectivity among Adaptor Proteins of Budding Yeast

Corresponding Author: Dr Maria Garcia Alai

Version 0:

Reviewer comments:

Reviewer #1

(Remarks to the Author)

In this manuscript, Defelipe et al. have reported the binding preferences and selectivity of adaptor proteins (APs) for clathrin in yeast. This research aimed to elucidate the evolutionary conservation and specialized roles of redundant APs in endocytosis and cellular trafficking. The findings offer molecular insights into AP selectivity, indicating that APs competitively bind clathrin and target different clathrin sites.

The authors have demonstrated the following information:

- 1) Using computational tools, the authors highlight that yeast adaptor proteins typically present bulky hydrophobic residues in CBMs, unlike their mammalian counterparts, which may account for differences in endocytic mechanisms.
- 2) Using thermal denaturation assays, the authors demonstrated that Ent5 has the highest affinity for clathrin.
- 3) The authors showed that while both Ent1 and Ent2 are important for endocytosis, they exhibit different binding patterns. By photobleaching FRET assay, they concluded that Ent1 shows stronger interactions with clathrin compared to Ent2.

Strengths:

The manuscript is well-written and easy to follow.

The protocols for protein purification and analysis follow gold standards in the field and are robust.

The use of experimental approaches such as the FRET photobleaching assay is well performed, though it requires additional controls for validation.

It is essential to use negative controls in this type of assay. Grow yeast expressing only mNeonGreen or only mScarlet. Measuring the fluorescence of yeast expressing only one of the tags ensures that donor fluorescence is not significantly affected by the photobleaching process.

Include yeast expressing a known interacting FRET pair tagged with mNeonGreen and mScarlet to validate the experimental setup might be also added.

Minor Comments

The title in Figure 2 should indicate that it represents the structural NTD from Chc for clarity.

The authors should explain the rationale behind choosing specific substitutions for clathrin (Q89A, K63E, I87D, and K98E), the Arrestin box (Q195A, K251E, I197T), and the W-Box (Q155A and F26A).

The study by Defelipe et al. provides significant insights into the molecular mechanisms of clathrin-mediated processes in yeast. The identification of binding preferences and the competitive nature of AP interactions with clathrin enhances our understanding of endocytosis and cellular trafficking. The experimental methodologies are robust.

Reviewer #2

(Remarks to the Author)

The study by Defelipe et al. provides important new insights into how the functionally critical NTD of Clathrin might utilize

multiple peptide binding sites to enable interactions with the adaptor protein (AP). The study features an impressive package of structure-function undertakings that appear to have been carried out rigorously and carefully.

I provide the following input for the authors' consideration.

What does the redundancy of binding of a single type of peptide to 3 distinct sites on the yeast Clathrin NTD suggest? Can all three sites be bound at the same time in vivo? What are the functional implications. There is an attempt in the Discussion section to consolidate these concepts/issues but it does not quite work well. For instance the authors state that their work 'provides molecular insights into the hierarchy and selectivity of adaptor proteins for clathrin binding...'. This is not clear to this reviewer. Please clarify.

Is there experimental evidence that the Clathrin NTD can bind a given peptide with a stoichiometry of $n=3$ in solution?

I would encourage the authors to introduce a creative new figure or annotate/expand an existing one (e.g. Figure 4) wherein the relative affinities are schematically mapped on a structural template. This would enhance the readability of the work given the multiplicity of the observed binding sites and the several structures.

Figure 4: Values for K_D in the table should not be in italics. Please use the correct nomenclature for K_D throughout the manuscript, i.e. K in italics and D as normal in the subscript!

Please provide a figure in the supplementary section featuring a panel of appropriate omit difference maps to provide evidence for the peptide occupancy per bound site.

Supplementary Table 3:

-R-merge is a fundamentally flawed crystallographic metric. Please report R_{meas} or R_{pim} , instead. In doing so please report values for overall and highest resolution shell.

-The reporting of $\langle I/\sigma(I) \rangle$ is ambiguous as stated. Please report values for overall and highest resolution shell.

-Report R/R_{free} consistently to 3 decimal places.

-Report average B -values for Protein, ligands, solvent. This is particularly important in this case given the centrality of the bound ligands in this work.

Please provide in the methods the molar ratios for each of the protein-peptide complexes crystallized.

In the section "Yeast Clathrin NTD complexes..." the sentence "The NTD part of the structures for all complexes are almost superimposable..." What do the authors mean by this? Please clarify and when applicable report rmsd values for these superpositions.

Supplementary Figure 4: It might be opportune to update these predictions via AlphaFold3.

What are the depicted predicted bound states representing? A representative of e.g. 5 very similar/identical predictions or are these singular models. In any case repeating with exercise with AF3 will generate 5 models per prediction run.

Reviewer #3

(Remarks to the Author)

Dr. Garcia-Alai and colleagues have presented an interesting work that provides molecular details on the selectivity of adaptor proteins for clathrin binding. The manuscript is very well structured, and I commend the authors for their thorough experimental plan.

The mass spectrometry experiments were meticulously designed, with comprehensive explanations of the calculations provided. I recommend that the inset in Supplementary Figure 6 a,b, and c be replaced with deconvoluted mass spectra for clarity, as this will enhance the visual representation of the data and aid in the understanding of the results (I defer this decision to the authors). Additionally, it would be advantageous for readers if the authors included a supplementary table with the intensity values used to determine the relative fraction of CHC bound to varying numbers of peptides.

Overall, a very thorough work! I recommend the publication of the manuscript.

Reviewer #4

(Remarks to the Author)

This manuscript reveals the interactions between clathrin-binding motif (CBM) from adaptor proteins related to endocytosis and the three different N-terminal domain (NTD) binding sites of clathrin heavy chain (CHC). Crystal structures of CBM motifs from different adaptor proteins in interaction with the 3 NTD sites were solved. Different binding affinities were determined by biophysical methods. Studies by FRET experiments and native mass-spectrometry reveal the preferences of epsins Ent1 and Ent2 CBM for different binding sites in the NTD domain of clathrin. In vivo observations of yeast cells by

TIRF microscopy show that Ent1-associated productive endocytic events are decreased in the Clathrin- and Arrestin-box double mutants cells.

In Figure 1, the author should indicate the length in aminoacids of the different proteins, indeed for some of them the CBM is at the C-terminus (example Ent1, Ent2), for some others it is located near the C-terminus (example Ent5), and some have their CBM in the middle of the protein (Sla1). This information is important, as not all CBM might be available for clathrin binding, especially for multiple sites clathrin binding.

In Figure 2, the authors present the crystal structure of the NTD in complex with the CBM from Ent1, Ent2, Ent5, Apl2 and Yap1801 (Fig.2). They should indicate (in the figure legend and the text) which of the Ent5 CBM sequence was used, as they are 2 CBM sequences that were identified in Ent5 (Figure 1b).

In the Methods chapter, it is indicated that the different CBM-Chc-NTD complexes were crystallized, however there are no indications on the method used to produce these complexes. Indeed, were the peptides and the proteins pre-incubated and the complex purified by size-exclusion chromatography (gel-filtration) or another methodology, or were the proteins and the peptides mixed and the crystals allowed to form?

There is also an important point about the ratio of CBM peptide/Chc-NTD protein, since in the crystal structure the 3 domains of Chc-NTD are occupied by CBM peptides, could this be due to an excess of CBM peptide compared to Chc-NTD used for the complex formation ?

Legend Figure 6, the peptide concentrations are ranging from 0 to 70 microM (not 50 as indicated).

For Figure 7, it is not clear on which bases the Abp- events versus the abortive events for Ent1 or Ent2 were classified? What are the differences between these two types of events? Indeed, based on the Figure 7a, the abortive event is shown as an event where Ent1 and Ent2 fluorescence intensity are increasing with time before decreasing, but this is not associated to a fluorescent signal for Abp1. However in the methods it is stated: "These events were further classified into Abp1 positive and Abp1 negative events using the dual color tracking functionality available in the package."

Moreover based on Figure 7d, most of the observed events even for the wild-type yeast cells are Abp1- negative meaning that they will not lead to a correct endocytic internalization event. Is this expected for this type of TIRF analysis?

Version 1:

Reviewer comments:

Reviewer #1

(Remarks to the Author)

I have reviewed the authors' responses and revisions to my comments. They have addressed all the concerns I raised, and I am satisfied with the changes. I believe the manuscript is now suitable for publication.

Reviewer #2

(Remarks to the Author)

The authors have addressed my remarks well and have prepared a substantially improved manuscript to set the stage for its publication in Nature Communications. This promises to be an important contribution in the field.

Reviewer #4

(Remarks to the Author)

The authors have answered to all the comments and the revised manuscript is of high quality.

REVIEWER COMMENTS

Reviewer #1 (Remarks to the Author):

In this manuscript, Defelipe et al. have reported the binding preferences and selectivity of adaptor proteins (APs) for clathrin in yeast. This research aimed to elucidate the evolutionary conservation and specialized roles of redundant APs in endocytosis and cellular trafficking. The findings offer molecular insights into AP selectivity, indicating that APs competitively bind clathrin and target different clathrin sites.

The authors have demonstrated the following information:

- 1) Using computational tools, the authors highlight that yeast adaptor proteins typically present bulky hydrophobic residues in CBMs, unlike their mammalian counterparts, which may account for differences in endocytic mechanisms.*
- 2) Using thermal denaturation assays, the authors demonstrated that Ent5 has the highest affinity for clathrin.*
- 3) The authors showed that while both Ent1 and Ent2 are important for endocytosis, they exhibit different binding patterns. By photobleaching FRET assay, they concluded that Ent1 shows stronger interactions with clathrin compared to Ent2.*

Strengths:

The manuscript is well-written and easy to follow.

The protocols for protein purification and analysis follow gold standards in the field and are robust.

The use of experimental approaches such as the FRET photobleaching assay is well performed, though it requires additional controls for validation.

It is essential to use negative controls in this type of assay. Grow yeast expressing only mNeonGreen or only mScarlet. Measuring the fluorescence of yeast expressing only one of the tags ensures that donor fluorescence is not significantly affected by the photobleaching process.

Response to reviewer:

As suggested by the reviewer, we conducted a control experiment to determine whether donor fluorescence is affected by the photobleaching process. Ent1-mNeonGreen-tagged cells were transformed with a vector expressing mScarlet under the CHC promoter (pRS315-mScarlet). Cells were treated with Latrunculin A to stall

endocytosis, allowing accurate measurement of endocytic events (puncta) before and after photobleaching. Our results demonstrate that photobleaching mScarlet by activating the 561 nm and 640 nm lasers does not significantly affect fluorescence intensity in the mNeonGreen channel. We have added Supplementary Figure 8 to the manuscript.

Supplementary Figure 8. FRET Specificity. Fluorescence intensity of Ent1-mNeonGreen-tagged cells excited with the 488 nm laser was measured before (Pre) and after photobleaching (Post) in the mScarlet channel (561 nm and 640 nm lasers). Data were analyzed using Yuen's trimmed means test¹, and no significant differences were observed, demonstrating that the photobleaching process does not affect donor fluorescence.

References

1. Yuen, K. K. The two-sample trimmed t for unequal population variances.

Biometrika **61**, 165–170 (1974).

In addition, the following text has been added to the relevant Results section:

“A control demonstrating that donor fluorescence is not affected by the photobleaching process is shown in Supplementary Fig. 8 (see the Methods section for a description of the experimental setup).”

and to the Methods section:

“As a FRET Photobleaching control, Ent1-mNeonGreen-tagged cells were transformed with a vector expressing mScarlet under the CHC promoter (pRS315-mScarlet). These cells were treated with Latrunculin A to stall endocytosis, allowing accurate measurement of endocytic events (puncta) before and after photobleaching.”

Include yeast expressing a known interacting FRET pair tagged with mNeonGreen and mScarlet to validate the experimental setup might be also added.

Response to reviewer:

We appreciate the reviewer's concern to include a known interacting FRET pair tagged with mNeonGreen and mScarlet to validate the experimental setup. We would like to clarify that mNeonGreen and mScarlet have already been established as effective FRET pairs in previous studies.^{1,2}

References

1. McCulloch, T. W., MacLean, D. M., & Kammermeier, P. J. (2020). Comparing the performance of mScarlet-I, mRuby3, and mCherry as FRET acceptors for mNeonGreen. *PloS one*, 15(2), e0219886.
2. Kohlmann, P., Krylov, S. N., Marchand, P., & Jose, J. (2024). FRET Assays for the Identification of *C. albicans* HSP90-Sba1 and Human HSP90 α -p23 Binding Inhibitors. *Pharmaceuticals*, 17(4), 516.

Minor Comments

The title in Figure 2 should indicate that it represents the structural NTD from Chc for clarity.

Response to reviewer:

We have corrected the figure caption according to the reviewer's suggestion.

“Figure 2. The same CBM motif binds to three different ChcNTD binding sites. Crystal structure of Chc-NTD (displayed as a gray cartoon) bound to CBMs from Ent1 (1.75 Å, red), Ent2 (1.74 Å, yellow), Ent5 (Ent5.1 LIDL motif, 2.01 Å, green), Apl2.1 (Apl2.1 LLDLF motif, 2.75 Å, blue) and Yap1801 (1.95Å, grey). All peptides bind to three binding boxes: Clathrin, Arrestin and W boxes. Zoom: individual Chc-NTD boxes with Ent2 CBM-containing peptide”

The authors should explain the rationale behind choosing specific substitutions for clathrin (Q89A, K63E, I87D, and K98E), the Arrestin box (Q195A, K251E, I197T), and the W-Box (Q155A and F26A).

Response to reviewer:

The substitutions were chosen following the criteria described below. We have added the following paragraph to the Methods section for clarification:

“Mutagenesis was performed according to the following criteria based on the chemical property of the amino acid: Lys/Arg residues are substituted with residues of opposite charge Asp/Glu. Polar residues are replaced by alanines and hydrophobic residues by Ala/Ser/Thr. In addition, we have selected combinations of mutants that could affect the interaction with targets but not compromise the thermal stability of the Chc-NTD fold, as predicted by FoldX 5¹ using a threshold of 1 kcal/mol.”

As an example, for the multiple mutant K63E, I87D, Q89A and K98E, the substitution I87D has a $\Delta\Delta G$ of 0.3 kcal/mol predicted by FoldX 5. Other substitutions like I87T, I87S, or I87A would further destabilize the fold ($\Delta\Delta G_{I87T} = 1.49$ kcal/mol, $\Delta\Delta G_{I87A} = 1.66$ kcal/mol, $\Delta\Delta G_{I87S} = 1.87$ kcal/mol).

Reference:

1. Delgado, Javier, et al. "FoldX 5.0: working with RNA, small molecules and a new graphical interface." *Bioinformatics* 35.20 (2019): 4168-4169.

We have changed the order of the residues to simplify the reading: e.g. Q89A, K63E, I87D, and K98E has now been replaced for K63E, I87D, Q89A and K98E. We have modified this in the text:

“We engineered a series of NTD mutants based on our structural analysis, aiming to disrupt the binding between the CBMs and the Chc boxes (see Methods). The specific mutants designed are as follows: for the Clathrin box, we introduced mutations K63E, I87D, Q89A and K98E (referred as the Cla mutant); for the Arrestin box, Q195A, I197T

and K251E (Arr mutant); and for the W-Box, we mutated residues F26A and Q155A (W mutant).”

The study by Defelipe et al. provides significant insights into the molecular mechanisms of clathrin-mediated processes in yeast. The identification of binding preferences and the competitive nature of AP interactions with clathrin enhances our understanding of endocytosis and cellular trafficking. The experimental methodologies are robust.

Reviewer #2 (Remarks to the Author):

The study by Defelipe et al. provides important new insights into how the functionally critical NTD of Clathrin might utilize multiple peptide binding sites to enable interactions with the adaptor protein (AP).

The study features an impressive package of structure-function undertakings that appear to have been carried out rigorously and carefully.

I provide the following input for the authors' consideration.

What does the redundancy of binding of a single type of peptide to 3 distinct sites on the yeast Clathrin NTD suggest? Can all three sites be bound at the same time in vivo? What are the functional implications. There is an attempt in the Discussion section to consolidate these concepts/issues but it does not quite work well. For instance the authors state that their work 'provides molecular insights into the hierarchy and selectivity of adaptor proteins for clathrin binding...'. This is not clear to this reviewer. Please clarify.

Response to reviewer:

We appreciate the reviewer's comments. It is, of course, not technically feasible to demonstrate that a single clathrin molecule is bound to three identical adaptors *in vivo*. However, our *in vitro* data and our mutants *in vivo* suggest that this could be a possibility. Our study indicates that multiple combinations are possible, and we provide evidence showing which adaptors exhibit tighter binding to NTD in a competitive scenario. We have now modified the discussion section to explain this more clearly.

“Given that several adaptors could compete for Chc binding sites, at similar endogenous concentrations, occupancy could be driven in a 'first come, first served' manner. However, our biophysical experiments indicate differences in dissociation constants across various CBM-containing peptides, with more than an order of

magnitude separating the strongest and weakest binders. This suggests a competitive scenario and, therefore, hierarchical binding in clathrin-mediated processes.”...

...“Our work provides molecular insights into a potential hierarchy and selectivity of adaptor proteins for clathrin binding. It shows that while these proteins can compete for binding sites, they have also evolved to target different sites, making it likely that a single NTD can be occupied by different adaptors.”...

...“The potential simultaneous binding of an adaptor to three sites would be an efficient mechanism for adaptor recruitment. The presence of multiple binding sites allows for specialization, as observed in our FRET experiments. Specifically, we observe a preference of Ent1 for the Clathrin box and of Ent2 for the Arrestin box. Our TIRF experiments provide further evidence supporting this model. When both the Cla and Arr binding sites were mutated, we observed a significant reduction in productive endocytic events.”

Is there experimental evidence that the Clathrin NTD can bind a given peptide with a stoichiometry of $n=3$ in solution?

Response to reviewer:

In 2015, Zhuo *et al.* used NMR and mapped chemical shifts on labeled bovine NTD upon binding to mouse AP180 peptides containing Clathrin-Box motifs. They could observe the occupancy of three regions corresponding to the Clathrin, Arrestin and W boxes. This experiment proved that a population of three sites can be occupied in solution. They have performed Isothermal Titration Calorimetry (ITC) experiments and the fitted data provided an $n=3$. We have also performed ITC experiments but were unable to reliably fit the data to any model with ‘ n ’ identical independent binding sites, most likely because the peptides we studied have different affinities for the binding sites (and possibly different enthalpies of binding). This is also supported by our native mass spectrometry experiments and the apparent binding affinities derived from nano differential scanning fluorimetry (nDSF).

Reference

Zhuo, Yue, et al. "Nuclear magnetic resonance structural mapping reveals promiscuous interactions between clathrin-box motif sequences and the N-terminal domain of the clathrin heavy chain." *Biochemistry* 54.16 (2015): 2571-2580.

This reference was added in the Introduction:

“NMR studies have revealed that a single peptide containing a Clathrin Binding Motif (CBM) is capable of binding to all three NTD boxes of bovine Clathrin²¹”

I would encourage the authors to introduce a creative new figure or annotate/expand an existing one (e.g. Figure 4) wherein the relative affinities are schematically mapped on a structural template. This would enhance the readability of the work given the multiplicity of the observed binding sites and the several structures.

Response to reviewer:

We appreciate the suggestion by the reviewer and have now included a new **Supplementary Figure 10**. Figure 4 shows global K_D s so these couldn't be mapped to single sites.

Supplementary Figure 10: Binding Affinities of Ent1 and Ent2 for WT Chc-NTD and Mutants. a) Determination of apparent dissociation constants (K_D^{APP}) by nano Differential Scanning Fluorimetry (nDSF) based on T_m shifts for Ent1 and Ent2 peptides with WT NTD and mutants at Clathrin (K63E, I87D, Q89A and K98E), Arrestin (Q195A, I197T and K251E), and W-box (F26A and Q155A) binding sites. b) Schematic

representation of these K_D^{App} values mapped onto the NTD structure (surface in gray). The presence of a peptide (Ent1 in red and Ent2 in yellow) indicates assumed peptide binding. Mutated residues are shown in red.

Figure 4: Values for KD in the table should not be in italics. Please use the correct nomenclature for KD throughout the manuscript, i.e. K in italics and D as normal in the subscript!

Response to reviewer:

We thank the reviewer for pointing this out, and we have corrected it in the revised version of the manuscript. The modified figure is included below.

Figure 4. Different APs show different binding affinities for NTD. a-f) Thermal denaturation first derivative curves followed at 350 nm display a thermal stabilization of the NTD upon binding to peptides containing CBMs observed by a shift of the T_m . g) K_D estimation for different CBMs using nDSF(see Methods). Colors in the sequence indicate type of amino acid: orange: hydrophobic; red: negatively charged; blue: polar. T_m : Melting temperature. CI95: Confidence Interval at 95%

Please provide a figure in the supplementary section featuring a panel of appropriate omit difference maps to provide evidence for the peptide occupancy per bound site.

Response to reviewer:

We have included a new supplementary figure (Supplementary Figure 2) depicting the density omit map at contour level of 3σ of each peptide in each site. The Supplementary Figure is included below.

Supplementary Figure 2. Omit map of each ScCHC-NTD/Peptide complex. The electron density map ($F_o - F_c$) surrounding these peptides is shown as a green mesh at a contour level of 3σ (standard deviation) above the mean electron density.

Supplementary Table 3:

-R-merge is a fundamentally flawed crystallographic metric. Please report R_{meas} or R_{pim} , instead. In doing so please report values for overall and highest resolution shell.

-The reporting of is ambiguous as stated. Please report values for overall and highest resolution shell.

-Report R/R_{free} consistently to 3 decimal places.

-Report average B-values for Protein, ligands, solvent. This is particularly important in this case given the centrality of the bound ligands in this work.

Response to reviewer:

We have corrected what was pointed out by the reviewer above. We have included the average B-factors for protein, peptide, ligands and water molecules. The corrected crystallographic data collection and refinement table is included below.

NTD Complex	Ent1	Ent2	Ent5	YAP1801	APL2
PDB ID	9EYT	9EXG	9EX5	9EXF	9EXT
Data collection					
Beamline	PETRAIII/P13	PETRAIII/P14	PETRAIII/P14	PETRAIII/P14	PETRAIII/P14
Space Group	P 43 21 2	P 21 21 21	C 2 2 21	P 1 21 1	P 41 21 2
Cell Dimensions					
a, b, c (Å)	56.31, 56.31, 254.60	46.50, 94.29, 269.03	77.00, 133.45, 285.53	51.08, 89.74, 188.12	141.78, 141.78, 168.60
α, β, γ (°)	90.0, 90.0, 90.0	90.0, 90.0, 90.0	90.0, 90.0, 90.0	90.0, 90.39, 90.0	90.0, 90.0, 90.0
Resolution (Å)	63.73 – 1.74 (1.77-1.74)	67.35 – 1.74 (1.77-1.74)	71.38 – 2.01 (2.05-2.01)	65.01 – 1.95 (1.98-1.95)	100.3 – 2.75 (2.85-2.75)
R_{pim}	0.019 (0.545)	0.037 (0.457)	0.049 (0.517)	0.063 (0.751)	0.022 (0.785)
$\langle I/\sigma \rangle$	27.2 (1.78)	16.7 (1.80)	15.1 (1.5)	11.0 (1.79)	30.0 (1.6)
CC 1/2	1 (0.843)	0.999 (0.788)	0.999 (0.533)	0.998 (0.553)	1 (0.589)
Completeness (%)	100 (99.9)	99.8 (99.8)	95.9 (34.9)	94.4 (86.3)	99.99 (100)
Redundancy	26 (27.5)	13.1 (12.9)	12.9(3.5)	7.7 (7.1)	13.1 (2.8)

Refinement					
No. of reflections (work/free)	43582/2114	121660/6253	93850/4607	116855/5916	45265/2259
Rwork/Rfree	0.185/0.230	0.191/0.225	0.182/0.214	0.183/0.214	0.228/0.258
Ramachandran favoured regions (%)	96.6	98.1	98.3	98.7	96.5
Ramachandran allowed regions (%)	3.4	1.9	1.7	1.2	3.5
Ramachandran outliers (%)	0	0	0	0.1	0.0
Rotamer outliers (%)	0.91	1.31	1.11	2.22	1.07
Clashscore	1.17	2.00	1.01	2.91	8.62
No. of atoms					
Protein	2844	8511	8452	11485	5698
Peptide	151	428	449	281	211
Ligands	0	5	0	1	0
Water	324	929	604	614	6
B-factors (Average)					
Protein	39.59	32.98	42.52	36.85	97.81
Peptide	48.48	37.71	51.28	50.99	114.20
Ligands	-	35.40	-	65.80	-
Water	43.43	35.53	45.74	36.51	92.60
RMS deviations					
Bond lengths (Å)	0.01	0.011	0.01	0.018	0.006
Bond angles (°)	1.7	1.57	1.45	1.83	0.93

Please provide in the methods the molar ratios for each of the protein-peptide complexes crystallized.

Response to reviewer:

We have now included the molar ratios for each of the protein-peptide complexes crystallized in the methods section. The modified text in the methods section is included below, highlighting the changes in red.

“Ent1 CBM/Chc-NTD complex was crystallized with a 2:1 (protein:precipitant) ratio in a 0.2 M potassium/sodium tartrate and 20 %(w/v) PEG 3350 solution. The protein

concentration was 10.4 mg/ml, while the peptide concentration was 10 mM (40:1 peptide:protein molar ratio). The first crystals appeared after one day and were harvested after 7 days. Ent2 complex was crystallized with a 2:1 (protein:precipitant) ratio in a 0.2 M lithium sulfate, 0.1 M Bis-Tris pH 5.5, 25%(w/v) PEG 3350 solution at 19°C. The protein concentration was 10.4 mg/ml, while the peptide concentration was 8 mM (30:1 peptide:protein molar ratio). The first crystals appeared after 3 days and were harvested after 7 days. Ent5.1 complex was crystallized with a 1:1 (protein:precipitant) ratio in a 10 % (w/v) PEG 1000, 10 % (w/v) PEG 8000 solution at 19 °C. The protein concentration was 9 mg/ml, while the peptide concentration was 5 mM (23:1 peptide:protein molar ratio). The first crystals appeared after 3 days and were harvested 84 days after setting up the plate. Apl2.1 complex was crystallized with a 1:2 ratio (protein:precipitant) in a 4 M sodium formate solution at 19°C. The protein concentration was 9 mg/ml, while the peptide concentration was 5 mM (23:1 peptide:protein molar ratio). Crystals appeared after 84 days and were harvested the following day. Yap1801 complex was crystallized with a 2:1 ratio (protein:precipitant) in a 0.1 M Tris-HCl pH 8.5, 0.28 M lithium sulfate monohydrate, 30% (w/v) PEG 4000 solution at 4°C. The protein concentration was 10 mg/ml, while the peptide concentration was 5 mM (20:1 peptide:protein molar ratio). Crystals appeared after 2 days and were harvested after 10 days. All complexes were cryoprotected using the respective mother liquor solution supplemented by 5% glycerol and peptide at the corresponding concentration.”

In the section “Yeast Clathrin NTD complexes...” the sentence “The NTD part of the structures for all complexes are almost superimposable...” What do the authors mean by this? Please clarify and when applicable report rmsd values for these superpositions.

Response to reviewer:

We calculated the Ca RMSD of the core of the domain (5-336 residues) and the differences to ScCHC-NTD-Ent2 are between 0.36 to 0.64 Å. The modified sentence is included below.

“The NTD structures for all complexes are very similar with Ca RMSDs between 0.36Å and 0.64Å (compared to Ent2 as reference): Ent1 = 0.64 Å, Ent5 = 0.57 Å, YAP1801 = 0.36 Å and APL2 = 0.53 Å)...”

Supplementary Figure 4: It might be opportune to update these predictions via AlphaFold3.

What are the depicted predicted bound states representing? A representative of e.g. 5 very similar/identical predictions or are these singular models. In any case repeating with exercise with AF3 will generate 5 models per prediction run.

Response to reviewer:

We appreciate the suggestion. Indeed, AF3 was released after submitting this paper. We have now repeated this for AF3. We included the following modifications (in red) in the Results section.

“To expand our current understanding, we utilized AlphaFold (AF2 Multimer V2.3 and AF3)^{1,2} to model other non-crystallized complexes of CBMs containing peptides with NTD. As a control, we compared our crystal structures with the predicted models (Supplementary Fig. 5 for AF2 and Supplementary Fig 6. for AF3 models). While for the Clathrin box, the predicted peptide position matches our experimental data, discrepancies are obtained for the Arrestin box, where AF2 failed to correctly place the Sla1 peptide (depicted within dashed red circles in Supplementary Fig 4.). In addition, we observe differences between our crystal structures of the W-box and the predicted complexes for Ent1, Gga2, Swa2 and Sla1. In these predictions, the peptides are not accurately positioned, leading to clashes. However, the AF2 model successfully predicts binding between the W-box and a second CBM found in Ent5 (Ent5.2). The substitutions we have observed between *S. cerevisiae* and mammalian species appear to influence their binding capabilities for the W-box. While incapable of binding WxxW peptides, the yeast W-box is capable of accommodating CBMs with a LLDW sequence.

For AF3 models (see Supplementary Fig. 6), we observe an improvement for the Sla1 and Swa2 complexes predictions. Peptides bind in the W-box as observed in the crystal structures. However, issues with the predictions for Ent1 and Gga2 in the W-box persist. Additionally, while AF2 Multimer correctly predicted Ent2 and Yap1802 in the W-box, AF3 produced incorrect models for these peptides.”

References

1. Evans, R. *et al.* Protein complex prediction with AlphaFold-Multimer. *bioRxiv* 2021.10.04.463034 (2022) doi:10.1101/2021.10.04.463034.
2. Abramson, J. *et al.* Accurate structure prediction of biomolecular interactions with AlphaFold 3. *Nature* **630**, 493–500 (2024).

Supplementary Figure 6. AlphaFold3 predictions for the NTD CBM complexes.
a) Superposition of the crystal structure of NTD bound to Ent1 peptide (cyan) and the

AF3 best model (blue). b) AF3 best predictions for NTD in complex with various CBM-containing peptides. Dashed red circles highlight wrong predictions in all five models. In addition the following paragraph was added in the Methods section:

“AF2.3 Multimer and AF3 Protein-peptide complex prediction.

We used AF2.3 Multimer (Evans et al., 2022) with default parameters. In the case of AF3 (Abramson et al., 2024), the AlphaFoldServer was used with one copy of Chc-NTD and three copies of each peptide for running the predictions. In all cases, the best model is shown in Supplementary Figs. 5 and 6. It is worth noting that when a poor prediction was detected, it was observed in all models.”

Reviewer #3 (Remarks to the Author):

Dr. Garcia-Alai and colleagues have presented an interesting work that provides molecular details on the selectivity of adaptor proteins for clathrin binding. The manuscript is very well structured, and I commend the authors for their thorough experimental plan.

The mass spectrometry experiments were meticulously designed, with comprehensive explanations of the calculations provided. I recommend that the inset in Supplementary Figure 6 a,b, and c be replaced with deconvoluted mass spectra for clarity, as this will enhance the visual representation of the data and aid in the understanding of the results (I defer this decision to the authors). Additionally, it would be advantageous for readers if the authors included a supplementary table with the intensity values used to determine the relative fraction of CHC bound to varying numbers of peptides.

Overall, a very thorough work! I recommend the publication of the manuscript.

Response to reviewer:

We have performed the suggested corrections by Reviewer#3 which are shown in the new Supplementary Figure 9 and Supplementary Tables 6 and 7.

Supplementary Figure 9 Native Mass Spectrometry spectra. a) Native MS spectra of free ScChc-NTD (black) and in presence of Ent1 at 5 μ M (green), 25 μ M (red) and 50 μ M (blue). Right, inset of the 11+ ions showing single, double and triple bound ScChc-NTD with Ent1. b) Native MS spectra of free ScChc-NTD (black) and in

presence of Ent2 at 5 μM (green), 25 μM (red) and 50 μM (blue). Right, inset of the 11+ ions showing single, double and triple bound ScCHC-NTD with Ent2. c) Native MS spectra of ScCHC-NTD in presence of both Ent1 and Ent2. Right, inset of the 11+ ions showing single, double and triple bound ScCHC-NTD with Ent1 and Ent2.

Reviewer #4 (Remarks to the Author):

This manuscript reveals the interactions between clathrin-binding motif (CBM) from adaptor proteins related to endocytosis and the three different N-terminal domain (NTD) binding sites of clathrin heavy chain (CHC). Crystal structures of CBM motifs from different adaptor proteins in interaction with the 3 NTD sites were solved. Different binding affinities were determined by biophysical methods. Studies by FRET experiments and native mass-spectrometry reveal the preferences of epsins Ent1 and Ent2 CBM for different binding sites in the NTD domain of clathrin. In vivo observations of yeast cells by TIRF microscopy show that Ent1-associated productive endocytic events are decreased in the Clathrin- and Arrestin-box double mutants cells.

In Figure 1, the author should indicate the length in aminoacids of the different proteins, indeed for some of them the CBM is at the C-terminus (example Ent1, Ent2), for some others it is located near the C-terminus (example Ent5), and some have their CBM in the middle of the protein (Sla1). This information is important, as not all CBM might be available for clathrin binding, especially for multiple sites clathrin binding.

Response to reviewer:

We appreciate the comment of the reviewer and have now clarified this in the text. In our study we have only selected those proteins that would contain CBMs which are predicted to be present in unstructured regions ($p\text{LDDT}<50$). The exposure of the region was checked against AF2 models deposited in UniProt. We clarified this in the methods section. We also added the total residue number between parentheses in Figure 1.

“Manual curation based on AlphaFold2 models deposited in UniProtKB was done for obtaining the final adaptor protein list, discarding proteins where the motif is in a structured region, only those regions with a low confidence and predicted as unstructured were considered ($p\text{LLTD}<50$).”

Figure 1. Discovery of a conserved clathrin binding motif in yeast. a) Comparison of mammalian and *S. cerevisiae* CBM motifs. In total, 28 hits in 24 proteins were found in the *Saccharomyces cerevisiae* genome, Φ denotes bulky hydrophobic residues b)

Alignment of CBM motifs from proteins related to endocytosis and trafficking. The numbers in parentheses represent the total protein length, while the numbers following the slash indicate the specific region being displayed. Dots following the protein name denote the sequence instance. c) Venn diagram of proteins involved in endocytosis (purple) and trafficking (pink) that contain the discovered motif.

In Figure 2, the authors present the crystal structure of the NTD in complex with the CBM from *Ent1*, *Ent2*, *Ent5*, *Apl2* and *Yap1801* (Fig.2). They should indicate (in the figure legend and the text) which of the *Ent5* CBM sequence was used, as they are 2 CBM sequences that were identified in *Ent5* (Figure 1b).

Response to reviewer:

We have now modified the figure following the reviewer's suggestion. We have clarified this for *Ent5.1* and *Apl2.1*. Below, we include the corrected Figure 2.

Figure 2. The same CBM motif binds to three different Chc NTD binding sites. Crystal structure of Chc-NTD (displayed as a gray cartoon) bound to CBMs from Ent1 (1.75 Å, red), Ent2 (1.74 Å, yellow), Ent5.1 (LIDL peptide, 2.01 Å, green), Apl2.1 (LLDLF peptide, 2.75 Å, blue) and Yap1801 (1.95Å, grey). All peptides bind to three binding boxes: Clathrin, Arrestin and W boxes. Zoom: individual Chc-NTD boxes with Ent2 CBM-containing peptide

In the Methods chapter, it is indicated that the different CBM-Chc-NTD complexes were crystallized, however there are no indications on the method used to produce these complexes. Indeed, were the peptides and the proteins pre-incubated and the complex purified by size-exclusion chromatography (gel-filtration) or another methodology, or were the proteins and the peptides mixed and the crystals allowed to form?

Response to reviewer:

We have made the following clarifications in the methods section of the revised version of the manuscript. All crystals were produced in SWISSCI plates by sitting drop. Protein and peptide were pre-mixed for co-crystallisation. The modified methods are included below.

“All the protein crystallization experiments were performed in 3-lens Swissci crystallization plates (SWISSCI, Zug, Switzerland) by sitting drop evaporation. NTD was incubated with excess peptide (between 23-40 fold) for 15 minutes at room temperature (21°C) prior to setting-up the crystal plates.”

There is also an important point about the ratio of CBM peptide/Chc-NTD protein, since in the crystal structure the 3 domains of Chc-NTD are occupied by CBM peptides, could this be due to an excess of CBM peptide compared to Chc-NTD used for the complex formation ?

Response to reviewer:

We agree with the reviewer that a high peptide concentration has been used for co-crystallisation. However, we also see this occupancy when performing native MS experiments with much lower peptide concentrations (up to 50 micromolar). In addition, our FRET experiments show that, in cellular concentrations, mutations to either the Clathrin or Arrestin boxes reduce the FRET efficiency.

Legend Figure 6, the peptide concentrations are ranging from 0 to 70 microM (not 50 as indicated).

Response to reviewer:

We appreciate this comment, and we have replaced the figure legend to include a clarification on the total peptide concentration used.

“Figure 6: One-site-occupancy binding analysis of Ent1 and Ent2 CBMs to Chc-NTD. a) Chc-NTD bound fraction in a titration of Ent1 peptide ranging from 0 to 50 μ M in the presence of 20 μ M Ent2 peptide (20 to 70 μ M total peptide concentration). b) Chc-NTD bound fraction in a titration of Ent2 peptide ranging from 0 to 50 μ M in the presence of 20 μ M Ent1 peptide (20 to 70 μ M total peptide concentration). Filled points represent the mean fraction of total Chc-NTD bound to the peptide, while empty points represent individual measurements. Error bars indicate the standard error (SE) of the mean.”

For Figure 7, it is not clear on which bases the Abp- events versus the abortive events for Ent1 or Ent2 were classified? What are the differences between these two types of events? Indeed, based on the Figure 7a, the abortive event is shown as an event where Ent1 and Ent2 fluorescence intensity are increasing with time before decreasing, but this is not associated to a fluorescent signal for Abp1. However in the methods it is stated: “These events were further classified into Abp1 positive and Abp1 negative events using the dual color tracking functionality available in the package.”

Response to reviewer:

We agree that the nomenclature used was confusing, so we have removed the terms "Abortive +/-" from the revised manuscript. This term has been used for mammalian endocytosis but is not relevant for yeast. We now focus our analysis solely on Abp1-positive and Abp1-negative events. The classification of events is based purely on whether Abp1 is recruited (Abp1+) or not (Abp1-) after Ent1/Ent2 using the dual-color tracking functionality available in the CMEAnalysis package. Below is the corrected methods section (marked in red).

“All Ent1/Ent2-positive events were categorized based on whether they could recruit Abp1 into the pit. Events were labeled as either Abp1+ (Abp1 recruited) or Abp1- (Abp1 not recruited). This classification was performed using the two-color tracking functionality available in CMEAnalysis.”

Moreover based on Figure 7d, most of the observed events even for the wild-type yeast cells are Abp1- negative meaning that they will not lead to a correct endocytic internalization event. Is this expected for this type of TIRF analysis?

Response to reviewer:

We appreciate the reviewer's comments and would like to clarify that we used TIRF microscopy because it allows us to accurately quantify the intensities and lifetimes of Ent1/2 signals at individual endocytic sites. Combined with the applied CMEAnalysis pipeline, this method provides statistically robust quantification of Abp1+/- endocytic events (see Fig. 7D). The fraction of Abp1- events could be related to the Abp1 signal occurring deeper within the cells and the strict intensity threshold used in the CMEAnalysis. For the Abp1+ events, our TIRF approach clearly indicates a significant decrease in frequency for the mutant Ent1 strains (Figure 7d and Supplementary Figure 11). This is not observed for Ent2. We have now tried to clarify these results in the text:

“Next, we compared the frequency of Ent1 or Ent2 at endocytic sites **which are Abp+**, between WT and Cla+Arr double mutant cells. This mutant has shown a lower binding *in vitro* affinity for both adaptors (Supplementary Fig. 10). The distribution plot of endocytic events for Ent1 shows a decrease in the percentage of Abp1+ sites (WT~33% vs. Cla+Arr ~15%). **This is supported by decreased Abp1 presence in combined NTD Cla mutants (Supplementary Fig. 11).** For Ent2, the total number of **Abp1+ sites is lower than those observed for Ent1 indicating its less frequent presence (Fig. 7d).** “